# BiPOLES is an optogenetic tool developed for bidirectional dual-color control of neurons

Johannes Vierock [1,8], Silvia Rodriguez-Rozada [2,8], Alexander Dieter[2], Florian Pieper [3], Ruth Sims[4], Federico Tenedini[5], Amelie C. F. Bergs [6], Imane Bendifallah[4], Fangmin Zhou [5], Nadja Zeitzschel[6], Joachim Ahlbeck[3], Sandra Augustin[1], Kathrin Sauter[2,5], Eirini Papagiakoumou [4], Alexander Gottschalk [6], Peter Soba [5,7], Valentina Emiliani [4], Andreas K. Engel[3], Peter Hegemann [1] & J. Simon Wiegert [2✉]

Optogenetic manipulation of neuronal activity through excitatory and inhibitory opsins has become an indispensable experimental strategy in neuroscience research. For many applications bidirectional control of neuronal activity allowing both excitation and inhibition of the same neurons in a single experiment is desired. This requires low spectral overlap between the excitatory and inhibitory opsin, matched photocurrent amplitudes and a fixed expression ratio. Moreover, independent activation of two distinct neuronal populations with different optogenetic actuators is still challenging due to blue-light sensitivity of all opsins. Here we report BiPOLES, an optogenetic tool for potent neuronal excitation and inhibition with light of two different wavelengths. BiPOLES enables sensitive, reliable dual-color neuronal spiking and silencing with single- or two-photon excitation, optical tuning of the membrane voltage, and independent optogenetic control of two neuronal populations using a second, blue-light sensitive opsin. The utility of BiPOLES is demonstrated in worms, flies, mice and ferrets.

[1] Institute for Biology, Experimental Biophysics, Humboldt University Berlin, Berlin, Germany. [2] Research Group Synaptic Wiring and Information Processing, Center for Molecular Neurobiology Hamburg, University Medical Center Hamburg-Eppendorf, Hamburg, Germany. [3] Department of Neurophysiology and Pathophysiology, University Medical Center Hamburg-Eppendorf, Hamburg, Germany. [4] Wavefront-Engineering Microscopy Group, Photonics Department, Institut de la Vision, Sorbonne Université, INSERM, CNRS, Institut de la Vision, Paris, France. [5] Research Group Neuronal Patterning and Connectivity, Center for Molecular Neurobiology Hamburg, University Medical Center Hamburg-Eppendorf, Hamburg, Germany. [6] Buchmann Institute for Molecular Life Sciences and Institute of Biophysical Chemistry, Goethe University, Frankfurt, Germany. [7] LIMES Institute, University of Bonn, Bonn, Germany. [8] These authors contributed equally: Johannes Vierock, Silvia Rodriguez-Rozada. ✉email: simon.wiegert@zmnh.uni-hamburg.de

To prove the necessity and sufficiency of a particular neuronal population for a specific behavior, a cognitive task, or a pathological condition, faithful activation, and inhibition of this population of neurons are required. In principle, optogenetic manipulations allow such interventions. However, excitation and inhibition of the neuronal population of interest are commonly done in separate experiments, where either an excitatory or inhibitory microbial opsin is expressed. Alternatively, if both opsins are co-expressed in the same cells, it is essential to achieve efficient membrane trafficking of both opsins, equal subcellular distributions, and a tightly controlled ratio between excitatory and inhibitory action at the specific wavelengths and membrane potentials, so that neuronal activation and silencing can be controlled precisely and predictably in all transduced cells. Precise co-localization of the two opsins is important when local, subcellular stimulation is required, or when control of individual neurons is intended, for example with two-photon holographic illumination[1]. Meeting these criteria is particularly challenging in vivo, where the optogenetic actuators are either expressed in transgenic lines or from viral vectors that are exogenously transduced. Ideally, both opsins are expressed from the same gene locus or delivered to the target neurons by a single viral vector. Moreover, for expression with fixed stoichiometry, the opsins should be encoded in a single open reading frame (ORF).

Previously, two strategies for stoichiometric expression of an inhibitory and an excitatory opsin from a single ORF were reported using either a gene fusion approach[2] or a 2A ribosomal skip sequence[3,4]. In both cases, a blue-light sensitive cation-conducting channel for excitation was combined with a red-shifted rhodopsin pump for inhibition. The gene fusion approach was used to systematically combine the inhibitory ion pumps halorhodopsin (NpHR), bacteriorhodopsin (BR), or archaerhodopsin (Arch) with a number of channelrhodopsin-2 (ChR2) mutants to generate single tandem-proteins[2]. While this strategy ensured co-localized expression of the inhibitory and excitatory opsins at a one-to-one ratio and provided important mechanistic insights into their relative ion-transport rates, membrane trafficking was not as efficient as with individually expressed opsins, thus limiting the potency of these fusion constructs for reliable control of neuronal activity.

The second strategy employed a 2A ribosomal skip sequence[3] to express the enhanced opsins ChR2(H134R)[5] and eNpHR3.0 as independent proteins at a fixed ratio from the same mRNA[4]. These bicistronic constructs, termed eNPAC, and eNPAC2.0[6], were used for bidirectional control of neuronal activity in various brain regions in mice[6–9]. While membrane trafficking of the individual opsins is more efficient compared to the gene fusion strategy, the expression ratio might still vary from cell to cell. Moreover, subcellular targeted co-localization (e.g., at the soma) is not easily achieved. Finally, functionality is limited in some model organisms such as *D. melanogaster*, since rhodopsin pumps are not efficient in these animals[10,11].

In addition to activation and inhibition of the same neurons, also independent optogenetic activation of two distinct neuronal populations is still challenging. Although two spectrally distinct opsins have been combined previously to spike two distinct sets of neurons[12–15], careful calibration and dosing of blue light were required to avoid activation of the red-shifted opsin. This typically leaves only a narrow spectral and energetic window to activate the blue-light but not the red-light-sensitive rhodopsin. Thus, dual-color control of neurons is particularly challenging in the mammalian brain where irradiance decreases by orders of magnitude over a few millimeters in a wavelength-dependent manner[16,17].

In order to overcome current limitations for bidirectional neuronal manipulations and to facilitate spiking of neuronal populations with orange-red light exclusively, in this work we systematically explore the generation of two-channel fusion proteins that combine red-light activated cation-channels and blue-light activated anion-channels enabling neuronal spiking and inhibition with red and blue light, respectively. With respect to previous bidirectional tools, inversion of the excitatory and inhibitory action spectra restricts depolarization to a narrow, orange-red spectral window since the inhibitory opsin compensates the blue-light-activated currents of the excitatory red-shifted channel. We show that among all tested variants, a combination of GtACR2[18] and Chrimson[12] termed BiPOLES (for Bidirectional Pair of Opsins for Light-induced Excitation and Silencing) proves most promising and allows (1) potent and reliable blue-light-mediated silencing and red-light-mediated spiking of pyramidal neurons in hippocampal slices; (2) bidirectional control of single neurons with single-photon illumination and two-photon holographic stimulation; (3) dual-color control of two distinct neuronal populations in combination with a second blue-light-sensitive ChR without cross-talk at light intensities spanning multiple orders of magnitude; (4) precise optical tuning of the membrane voltage between the chloride and cation reversal potentials; (5) bidirectional manipulations of neuronal activity in a wide range of invertebrate and vertebrate model organisms including worms, fruit flies, mice, and ferrets.

## Results

**Engineering of BiPOLES and biophysical characterization in HEK cells.** To identify suitable combinations of opsins for potent membrane voltage shunting or depolarization with blue and red light, respectively, we combined the blue-light or green-light sensitive anion-conducting channelrhodopsins (ACRs) Aurora[11], iC++[19], GtACR1, and GtACR2[18] with the red-light sensitive cation-conducting channelrhodopsin (CCR) Chrimson[12]; or conversely, the blue-light sensitive GtACR2 with the red-light sensitive CCRs bReaChES[20], f-Chrimson, vf-Chrimson[21], and ChRmine[22] (Fig. 1a). We fused these opsin-pairs with different linkers, expanding previous rhodopsin fusion strategies[2,23] to obtain optimal expression and membrane targeting. The linkers were composed of the Kir2.1 membrane trafficking signal (TS)[4], different arrangements of a cyan or yellow fluorescent protein, and the transmembrane β helix of the rat gastric $H^+/K^+$ ATPase (βHK) to maintain the correct membrane topology of both opsins[2] (Fig. 1a).

For a detailed biophysical evaluation, we expressed all ACR-CCR tandems in human embryonic kidney (HEK) cells and recorded blue-light and red-light evoked photocurrents in the presence of a chloride gradient. In all constructs, except the one lacking the βHK-subunit (L3, Fig. 1a), blue-light-activated currents were shifted towards the chloride Nernst potential whereas red-light-activated currents were shifted towards the Nernst potential for protons and sodium (Fig. 1b–d, Supplementary Fig. 1), indicating functional membrane insertion of both channels constituting the tandem constructs. Reversal potentials (Fig. 1d) and photocurrent densities (Fig. 1e) varied strongly for the different tandem variants indicating considerable differences in their wavelength-specific anion/cation conductance ratio and their membrane expression. Photocurrent densities were not only dependent on the identity of the fused channels, but also on the sequence of both opsins in the fusion construct, as well as the employed fusion linker. In contrast to a previous study[2], the optimized linker used in this study did not require a fluorescent protein to preserve the functionality of both channels (L4, Fig. 1a, d, e). Direct comparison of red-light and blue-light evoked photocurrent densities with those of βHK-Chrimson and GtACR2 expressed alone indicated that most tandem constructs

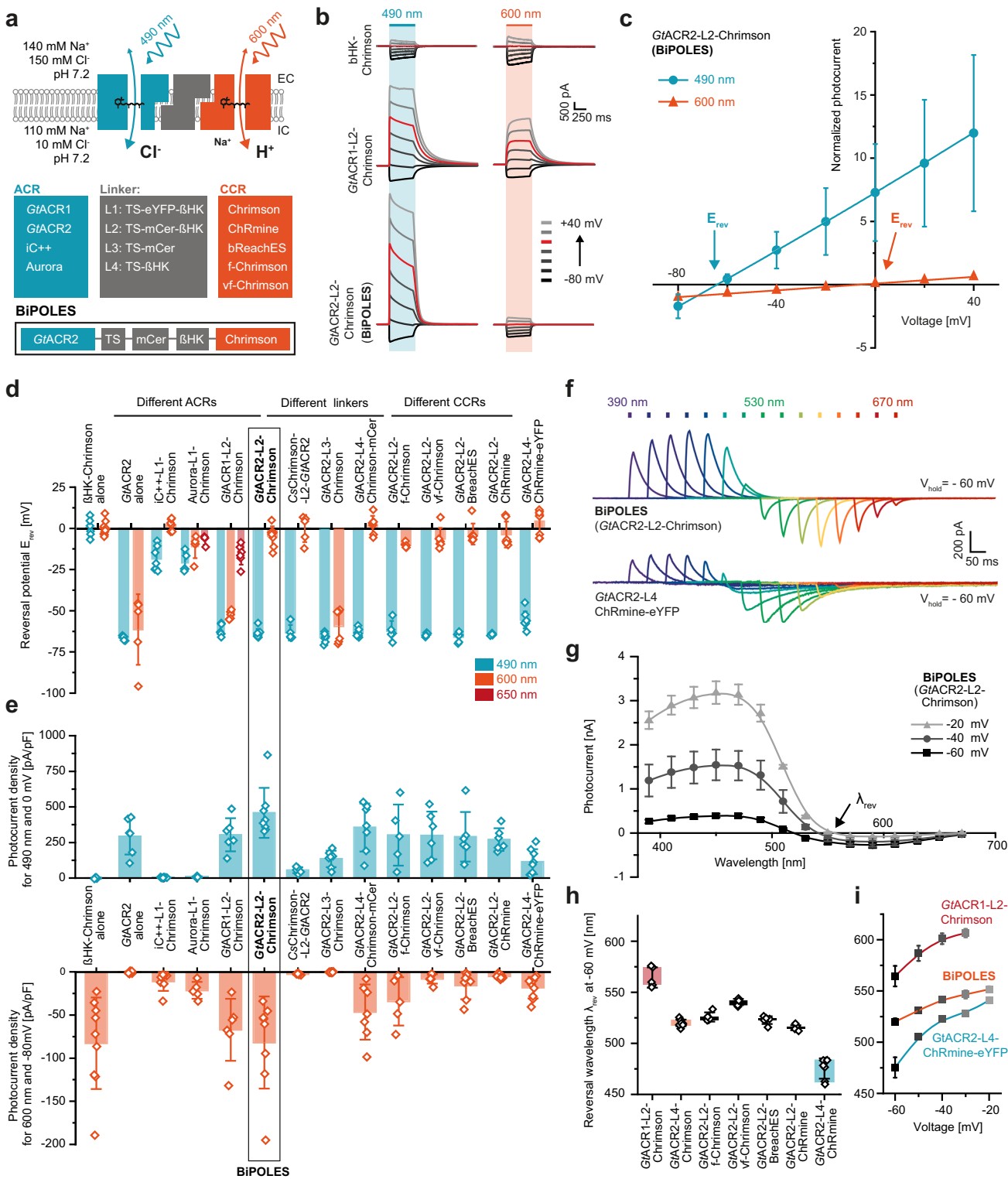

harboring a *Gt*ACR reached similar membrane expression efficacy as the individually expressed channels (Fig. 1e).

At membrane potentials between the Nernst potentials for chloride and protons, blue and red light induced outward and inward currents, respectively, in all *Gt*ACR-fusion constructs. (Fig. 1e–g, Supplementary Fig. 1). The specific wavelength of photocurrent inversion ($\lambda_{rev}$) was dependent on the absorption spectra and relative conductance of the employed channels, as

well as on the relative ionic driving forces defined by the membrane voltage and the respective ion gradients (Fig. 1g–i). The red-shift of $\lambda_{rev}$ for the vf-Chrimson tandem compared to BiPOLES reflects the reduced conductance of this Chrimson mutant (Fig. 1h, Supplementary Fig. 1c), as already previously shown[21,24], whereas the blue-shift of $\lambda_{rev}$ for the ChRmine tandem with L4 (Fig. 1f, h) is explained by the blue-shifted activation spectrum of ChRmine compared to Chrimson[25] and its

**Fig. 1 Development of BiPOLES and biophysical characterization. a** Molecular scheme of BiPOLES with the extracellular (EC) and intracellular (IC) ionic conditions used for HEK293-cell recordings. The blue-green-light-activated natural anion channels GtACR1 and GtACR2 or the engineered ChR-chimeras iC ++ and Aurora were fused to the red-light-activated cation channels Chrimson, ChRmine, bReaChES, f-Chrimson, or vf-Chrimson by different linker regions consisting of a trafficking signal (ts), a yellow or cyan fluorescent protein (eYFP, mCerulean3) and the βHK transmembrane fragment. The fusion construct termed BiPOLES is indicated by a black frame. **b** Representative photocurrents of βHK-Chrimson-mCerulean (top), GtACR1-ts-mCerulean-βHK-Chrimson (middle) GtACR2-ts-mCerulean-βHK-Chrimson (BiPOLES, bottom) in whole-cell patch-clamp recordings from HEK293 cells at 490 nm and 600 nm illumination. **c** Normalized peak photocurrents of BiPOLES at different membrane voltages evoked at either 490 or 600 nm (see panel **b**, mean ± SD; n = 8 independent cells; normalized to the peak photocurrent at −80 mV and 600 nm illumination). **d** Reversal potential of peak photocurrents during 500-ms illumination with 490, 600, or 650 nm light as shown in **b** (mean ± SD). **e** Peak photocurrent densities for 490 nm and 600 nm excitation at 0 mV (close to the reversal potential of protons and cations) and −80 mV (close to the reversal potential for chloride) measured as shown in **b** (mean ± SD; for both **d** and **e** n = 5 biological independent cells for Aurora-L1-Chrimson, CsChrimson-L2-GtACR2 and GtACR2-L2-f-Chrimson; n = 6 for GtACR2, GtACR1-L2-Chrimson and GtACR2-L2-vf-Chrimson; n = 7 for iC++-L1-Chrimson, GtACR2-L3-Chrimson, GtACR2-L4-Chrimson-mCer, GtACR2-L2-BreachES, and GtACR2-L2-ChRmine; n = 8 for GtACR2-L2-Chrimson and n = 9 for ßHK-Chrimson and GtACR2-L4-ChRmine-ts-eYFP-er). **f** Representative photocurrents of BiPOLES (top) and GtACR2-L4-ChRmine-ts-eYFP-er (bottom) with 10 ms light pulses at indicated wavelengths and equal photon flux at −60 mV. **g** Action spectra of BiPOLES at different membrane voltages (λ_rev = photocurrent reversal wavelength, mean ± SEM, n = 9 independent cells for −60 mV, n = 4 for −40 mV and n = 2 for −20 mV). **h** Photocurrent reversal wavelength λ_rev at −60 mV (mean ± SD, n = 5 independent cells for GtACR1-L2-Chrimson and GtACR2-L2-f-Chrimson, n = 6 for GtACR2-L2-vf-Chrimson and GtACR2-L2-ChRmine, n = 7 for GtACR2-L4-ChRmine-ts-eYFP-er, n = 8 for GtACR2-L2-BreachES and n = 9 for GtACR2-L2-Chrimson). **i** λ_rev of GtACR1-L2-Chrimson, BiPOLES, and GtACR2-L4-ChRmine-TS-eYFP-ER at different membrane voltages (mean ± SD; n = 5 biological independent cells for GtACR1-L2-Chrimson, n = 7 for GtACR2-L4-ChRmine-ts-eYFP-er and n = 9 for GtACR2-L2-Chrimson).

presumably large single-channel conductance. Switching the L4 linker to L2 shifted λ_rev to longer wavelengths for the ChRmine fusion constructs at the expense of ChRmine photocurrents (Fig. 1e, h), pointing to a stronger impact of the protein linker on the ChRmine photocurrent compared to other red-shifted CCRs (Fig. 1e).

Among all tested combinations, GtACR2-L2-Chrimson—from here on termed BiPOLES—was the most promising variant. First, it showed the largest photocurrent densities of all tested fusion constructs (Fig. 1e,f), second, reversal potentials for blue or red light excitation were close to those of individually expressed channels (−64 ± 3 mV and −5 ± 6 mV for BiPOLES compared to −66 ± 2 mV and 0 ± 5 mV of GtACR2 and βHK-Chrimson expressed alone, Fig. 1c, d, Supplementary Fig. 1b) and third, peak activity of the inhibitory anion and excitatory cation current had the largest spectral separation among all tested variants (150 ± 5 nm, Fig. 1f, g). Thus, BiPOLES enables selective activation of large anion and cation currents with spectrally well-separated wavelengths (Fig. 1e). BiPOLES was remarkably better expressed in HEK-cells than the previously reported ChR2-L1-NpHR fusion construct[2] and featured larger photocurrents at −60 mV than the bicistronic construct eNPAC2.0[6] (Supplementary Fig. 2a–c). Moreover, employing an anion channel with high conductance instead of a chloride pump, which transports one charge per absorbed photon and is weak at a negative voltage, yielded chloride currents in BiPOLES expressing cells at irradiances 2 orders of magnitude lower than with eNPAC2.0 (Supplementary Fig. 2d–f). Anion conductance in BiPOLES was sufficiently large to compensate inward currents of Chrimson even at high irradiance, driving the cell back to the chloride Nernst potential, which is close to the resting membrane voltage (Supplementary Fig. 2d–f). We further verified the implementation of an anion-conducting channel by testing whether sufficient blue-light hyperpolarization could be achieved with a rhodopsin pump[26] instead of a channel. Replacing GtACR2 with a blue-light sensitive proton pump led to barely detectable outward currents at the same irradiance due to low ion turnover of the ion pump under the given voltage and ion conditions (Supplementary Fig. 2d, g).

**Evaluation of BiPOLES in CA1 pyramidal neurons.** Next, we validated BiPOLES as an optogenetic tool for bidirectional control of neuronal activity. In CA1 pyramidal neurons of rat hippocampal slice cultures, illumination triggered photocurrents with biophysical properties similar to those observed in HEK cells (Fig. 2a, b, Supplementary Fig. 3a–c). We observed membrane-localized BiPOLES expression most strongly in the somatodendritic compartment (Fig. 2c, Supplementary Fig. 3d). However, some fraction of the protein accumulated inside the cell in the periphery of the cell nucleus, indicating sub-optimal membrane trafficking of BiPOLES. To enhance membrane trafficking, we generated a soma-targeted variant (somBiPOLES) by attaching a C-terminal Kv2.1-trafficking sequence[27]. Soma targeting has the additional benefit of avoiding the expression of the construct in axon terminals, where the functionality of BiPOLES might be limited due to an excitatory chloride reversal potential and subsequent depolarizing action of GtACR2[28,29]. somBiPOLES showed strongly improved membrane localization restricted to the cell soma and proximal dendrites with no detectable intracellular accumulations (Fig. 2c, Supplementary Fig. 3d). Compared to BiPOLES, blue-light and red-light mediated photocurrents were enhanced and now similar in magnitude to those in neurons expressing either Chrimson or soma-targeted GtACR2 (somGtACR2), alone (Fig. 2d, Supplementary Fig. 4a, 5a, b). Passive and active membrane parameters of BiPOLES-expressing and somBiPOLES-expressing neurons were similar to non-transduced, wild-type neurons (Supplementary Fig. 6), indicative of good tolerability in neurons.

To verify the confinement of somBiPOLES to the somatodendritic compartment despite the improved expression, we virally transduced area CA3 in hippocampal slice cultures with somBiPOLES and recorded optically evoked EPSCs in postsynaptic CA1 cells. Local illumination with red light in CA3 triggered large excitatory postsynaptic currents (EPSCs), while local red illumination of axon terminals in CA1 (635 nm, 2 pulses of 5 ms, 40 ms ISI, 50 mW mm−2), did not trigger synaptic release, indicating the absence of somBiPOLES from axonal terminals (Supplementary Fig. 3e,f). Thus, despite enhanced membrane trafficking, somBiPOLES remained confined to the somatodendritic compartment.

Having shown that somBiPOLES is efficiently expressed in CA1 pyramidal cells, we next systematically benchmarked light-evoked spiking and inhibition parameters for somBiPOLES by direct comparison to Chrimson or somGtACR2 expressed in

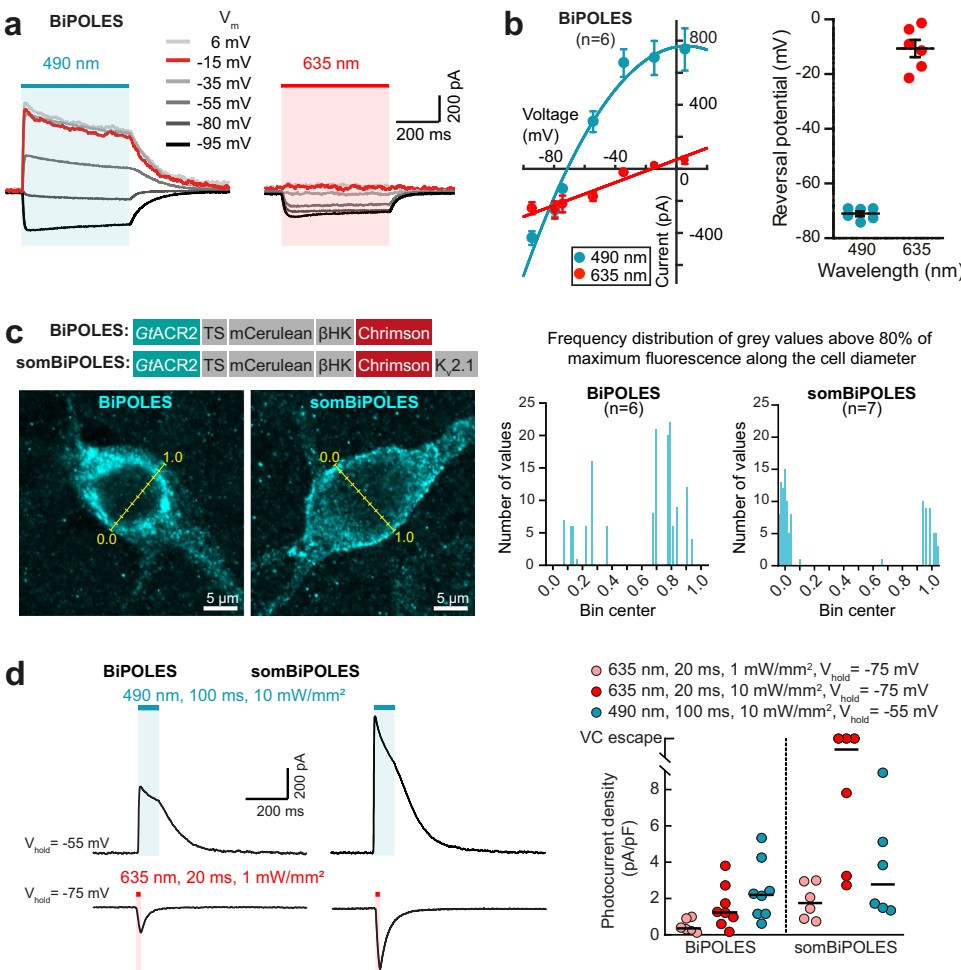

**Fig. 2 Expression and functional characterization of BiPOLES and somBiPOLES in hippocampal neurons. a** Representative photocurrent traces of BiPOLES in CA1 pyramidal neurons at indicated membrane voltages ($V_m$: from −95 to +6 mV) upon illumination with 490 or 635 nm (500 ms, 10 mW mm$^{-2}$). **b** Left: quantification of photocurrent-voltage relationship (symbols: mean ± SEM, $n = 6$ cells, lines: polynomial regression fitting, $R^2 = 0.98$ and 0.94, for 490 and 635 nm, respectively). Right: reversal potential under 490 or 635 nm illumination (black lines: mean ± SEM, $n = 6$ cells). **c** Left: Molecular scheme of BiPOLES and somBiPOLES as used in neurons. Representative maximum-intensity projection images of immunostainings showing expression of BiPOLES or soma-targeted BiPOLES (somBiPOLES) in CA3 pyramidal neurons of organotypic hippocampal slices. Yellow lines indicate the bins used to measure fluorescence intensity along the cell equator. Right: Frequency distribution of gray values above 80% of the maximum fluorescence intensity measured along the cell diameter in BiPOLES-expressing ($n = 6$ cells) and somBiPOLES-expressing CA3 cells ($n = 7$ cells). Note improved trafficking of somBiPOLES to the cell membrane, shown by the preferential distribution of brighter pixels around bins 0.0 and 1.0. **d** Left: Representative photocurrent traces measured in BiPOLES-expressing or somBiPOLES-expressing CA1 pyramidal neurons. Inward cationic photocurrents evoked by a 635 nm light pulse (20 ms, 1 mW mm$^{-2}$) were recorded at a membrane voltage of −75 mV, and outward anionic photocurrents evoked by a 490 nm light pulse (100 ms, 10 mW mm$^{-2}$) were recorded at a membrane voltage of −55 mV. Right: Quantification of photocurrent densities evoked under the indicated conditions. Note that photocurrent densities were strongly enhanced for somBiPOLES compared to BiPOLES (black horizontal lines: medians, $n_{BiPOLES} = 8$ cells, $n_{somBiPOLES} = 6$ cells.

hippocampal CA1 pyramidal neurons, respectively (Fig. 3, Supplementary Figs. 4, 5). To compare spiking performance in somBiPOLES or Chrimson expressing CA1 pyramidal cells, we delivered trains of 5-ms blue (470 nm), orange (595 nm), or red (635 nm) light pulses at irradiances ranging from 0.1 to 100 mW mm$^{-2}$. Action potential (AP) probability in somBiPOLES neurons reached 100% at 0.5 mW mm$^{-2}$ with 595 nm and 10 mW mm$^{-2}$ with 635 nm light, similar to neurons expressing Chrimson alone (Fig. 3b,c). In pyramidal cells, action potentials (APs) could be reliably driven up to 10-20 Hz with somBiPOLES (Supplementary Fig. 7c) similar to Chrimson alone, as shown previously[12]. Delivering the same number of photons in a time range of 1–25 ms did not alter the AP probability, but longer pulses increased sub-threshold depolarization (Supplementary Fig. 7d).

In contrast to orange or red light, blue light did not evoke APs at any irradiance in somBiPOLES neurons due to the activity of the blue-light sensitive anion channel. On the contrary, neurons expressing Chrimson alone reached 100% AP firing probability at 10 mW mm$^{-2}$ with 470 nm (Fig. 3b, c). Using light ramps with gradually increasing irradiance enabled us to precisely determine the AP threshold and to quantitatively compare the spiking efficacy of different excitatory opsins. The irradiance threshold for the first AP was similar for somBiPOLES and Chrimson at 595 nm ($0.74 ± 0.06$ mW mm$^{-2}$ for somBiPOLES and $0.68 ± 0.05$ mW mm$^{-2}$ for Chrimson) reflecting that the functional expression levels were similar. In contrast, blue light triggered APs at $0.95 ± 0.09$ mW mm$^{-2}$ in Chrimson expressing cells, but never in somBiPOLES or BiPOLES neurons (Fig. 3d, e, Supplementary Fig. 7a, b). Thus, somBiPOLES enables neuronal excitation

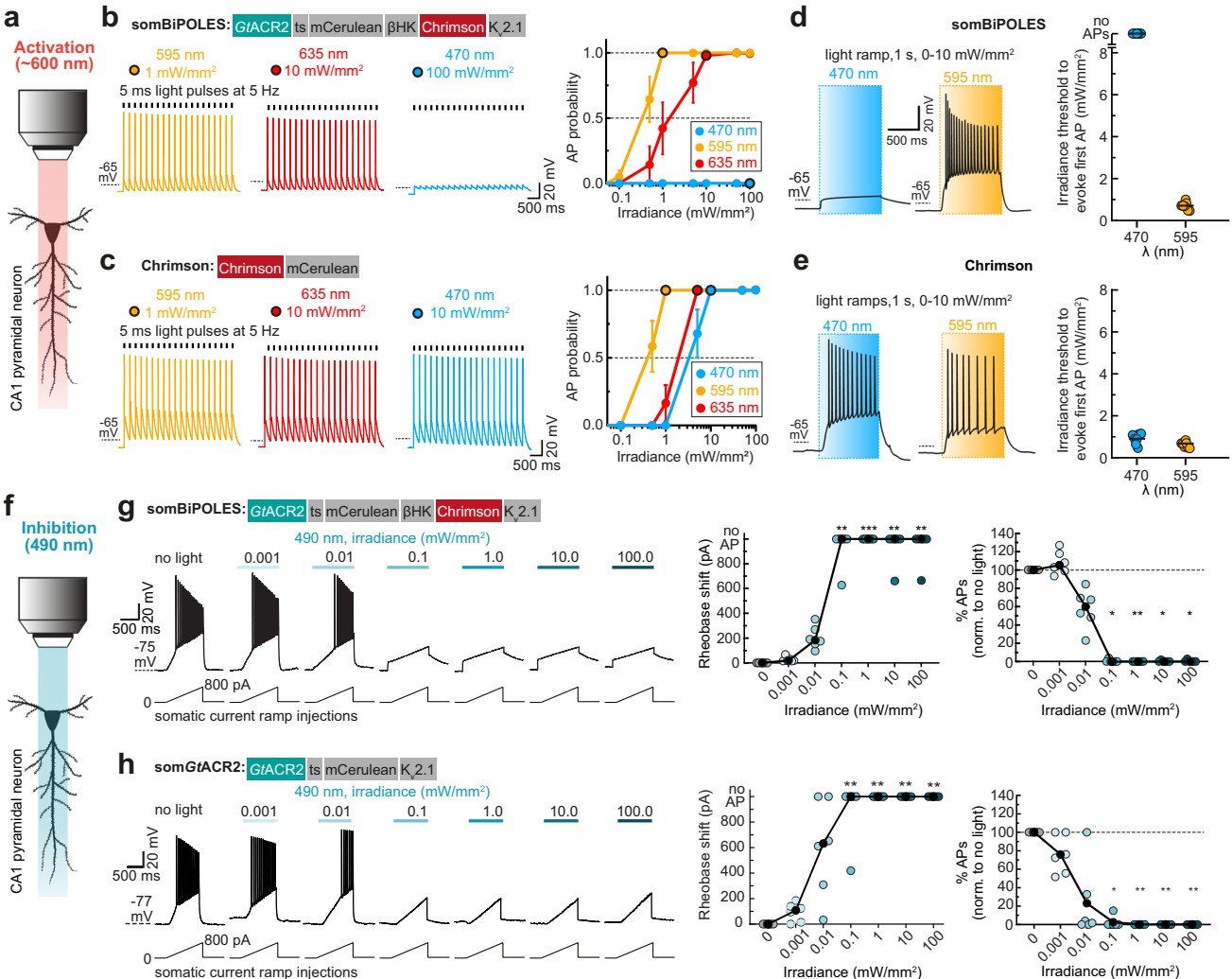

**Fig. 3 somBiPOLES allows potent dual-color spiking and silencing of the same neurons using red and blue light, respectively. a** Quantification of neuronal excitation with somBiPOLES or Chrimson only. **b** Optical excitation is restricted exclusively to the orange/red spectrum in somBiPOLES-expressing neurons. Left: Example traces of current-clamp (IC) recordings in somBiPOLES-expressing CA1 pyramidal cells to determine light-evoked action potential (AP)-probability at different wavelengths. Right: quantification of light-mediated AP probability at indicated wavelengths and irradiances (symbols correspond to mean ± SEM, $n = 8$ cells). Black outlined circles correspond to irradiance values shown in example traces on the left. **c** Same experiment as shown in **b**, except that CA1 neurons express Chrimson only (symbols correspond to mean ± SEM, $n = 7$ cells) Note blue-light excitation of Chrimson, but not somBiPOLES cells. **d** Light-ramp stimulation to determine the AP threshold irradiance. Left: Representative membrane voltage traces measured in somBiPOLES-expressing CA1 pyramidal neurons. The light was ramped linearly from 0 to 10 mW mm$^{-2}$ over 1s. Right: Quantification of the irradiance threshold at which the first AP was evoked (black horizontal lines: medians, $n = 7$ cells). **e** Same experiment as shown in (**b**), except that CA1 neurons express Chrimson only (black horizontal lines: medians, $n = 7$ cells). The threshold for action potential firing with 595 nm was similar between somBiPOLES-expressing and Chrimson-expressing neurons, while somBiPOLES cells were not sensitive to blue light. **f** Quantification of neuronal silencing with somBiPOLES or som$Gt$ACR2 only. **g** somBiPOLES mediates neuronal silencing upon illumination with blue light. Left: Current ramps (from 0–100 to 0–900 pA) were injected into somBiPOLES-expressing CA1 pyramidal cells to induce APs during illumination with blue light at indicated intensities (from 0.001 to 100 mW mm$^{-2}$). The injected current at the time of the first action potential was defined as the rheobase. Right: Quantification of the rheobase shift and the relative change in the number of ramp-evoked action potentials. Illumination with 490 nm light of increasing intensities activated somBiPOLES-mediated Cl$^-$ currents shifting the rheobase to higher values and shunting action potentials. **h** Same experiment is shown in **g**, except that CA1 neurons express som$Gt$ACR2 only. Note similar silencing performance of somBiPOLES and $Gt$ACR2. In **h**, **g** black circles correspond to medians, $n_{\text{somBiPOLES}} = 6$ cells, $n_{\text{som}Gt\text{ACR2}} = 6$ cells, one-way Friedman test, *$p < 0.05$, **$p < 0.01$, ***$p < 0.001$.

exclusively within a narrow spectral window restricted to orange-red light, avoiding inadvertent blue-light mediated spiking.

Next, we quantified the silencing capacity of somBiPOLES and compared it to som$Gt$ACR2 alone—the most potent opsin for blue-light mediated somatic silencing[28,29]—by measuring the capacity to shift the threshold for electrically evoked APs (i.e., rheobase, see "Methods" section). Both variants similarly shifted the rheobase towards larger currents starting at an irradiance of

0.1 mW mm$^{-2}$ with 490 nm light, leading to a complete block of APs in most cases (Fig. 3g, h). Neuronal silencing was efficient under 490 nm-illumination, even at high irradiances (up to 100 mW mm$^{-2}$, Fig. 3g), showing that blue light cross-activation of Chrimson in somBiPOLES did not compromise neuronal shunting.

We compared somBiPOLES with eNPAC2.0, the most advanced optogenetic tool currently available for dual-color

excitation and inhibition[4,6,7]. In eNPAC2.0 expressing CA1 pyramidal neurons, depolarizing and hyperpolarizing photocurrents were present under blue and yellow/orange light, respectively (Supplementary Fig. 8a), consistent with its inverted action spectrum compared to BiPOLES (Supplementary Fig. 2). Compared to BiPOLES (Supplementary Fig. 3c) peak photocurrent ratios were more variable between cells (Supplementary Fig. 8a), indicative of different stoichiometries between ChR2(HR) and eNpHR3.0 in different neurons, probably because membrane trafficking and degradation of both opsins occur independently. Moreover, blue-light-evoked spiking with eNPAC2.0 required approx. 10-fold higher irradiance compared to somBiPOLES and did not reach 100% reliability (Supplementary Fig. 8c), which might be explained by cross-activation of eNpHR3.0 under high blue irradiance (see also Supplementary Fig. 2d). Blue-light-triggered APs could not be reliably blocked with concomitant yellow illumination at 10 mW mm$^{-2}$ (Supplementary Fig. 8b). Further on, activation of eNPAC2.0 (i.e., eNpHR3.0) with yellow light (580 nm) caused strong membrane hyperpolarization followed by rebound spikes in some cases (Supplementary Fig. 8d). Finally, and consistent with photocurrent measurements in HEK cells (Supplementary Fig. 2e, f), silencing of electrically evoked APs required 100-fold higher irradiance with eNPAC2.0, compared to somBiPOLES, until a significant rheobase-shift was observed (Supplementary Fig. 8e).

In summary, somBiPOLES is suitable for potent, reliable neuronal activation exclusively with orange-red light and silencing with blue light. somBiPOLES displays similar potency for neuronal excitation and inhibition as Chrimson and som*Gt*ACR2 alone.

**BiPOLES allows various neuronal manipulations with visible light**. We evaluated BiPOLES and somBiPOLES in the context of three distinct neuronal applications: bidirectional control of neuronal activity, optical tuning of the membrane voltage, and independent spiking of two distinct neuronal populations.

We first tested the suitability of BiPOLES and somBiPOLES for all-optical excitation and inhibition of the same neurons (Fig. 4a). Red light pulses (635 nm, 20 ms, 10 mW mm$^{-2}$) reliably triggered APs in somBiPOLES expressing neurons (Fig. 4b), while APs were triggered only in approx. 50% of BiPOLES expressing neurons under these stimulation conditions (Supplementary Fig. 7e), due to a higher irradiance threshold to evoke APs in those cells (Supplementary Fig. 7a, b). Concomitant blue illumination (490 nm, 10 mW mm$^{-2}$) for 100 ms reliably blocked red-light evoked APs in all cases. As expected from an anion conducting channel, blue light alone had only a minor impact on the resting membrane voltage, due to the close proximity of the chloride reversal potential to the resting potential of the cell (Fig. 4b, Supplementary Fig. 7e) In contrast, neurons expressing Chrimson alone showed APs both under red and blue illumination (Supplementary Fig. 4b).

Aside from dual-color spiking and inhibition, a major advantage of the fixed 1:1 stoichiometry between an anion and cation channel with different activation spectra in BiPOLES is the ability to precisely tune the ratio between anion-conductance and cation-conductance with light (Fig. 1f,g, Supplementary Fig. 3c). In neurons, this allows to optically tune the membrane voltage between the chloride reversal potential and the action potential threshold (Fig. 4c). Optical membrane voltage tuning was achieved either by a variable ratio of blue and orange light at the absorption peak wavelengths of *Gt*ACR2 and Chrimson (Fig. 4d) or by using a single color with fixed irradiance over a wide spectral range (Fig. 4e). Both approaches yielded reliable and reproducible membrane voltage shifts. Starting from the chloride

Nernst potential when only *Gt*ACR2 was activated with blue light at 470 nm, the membrane depolarized steadily with an increasing 595/470 nm ratio, eventually passing the action potential threshold (Fig. 4d). Similarly, tuning a single wavelength between 385 nm and 490 nm clamped the cell near the Nernst potential for chloride, while shifting the wavelength peak further towards red led to gradual depolarization, eventually triggering action potentials at 580 nm (Fig. 4e). Depending on the available light source both methods allow precise control of anion and cation fluxes at a fixed ratio and might be applied for locally defined subthreshold membrane depolarization in single neurons or to control the excitability of networks of defined neuronal populations.

Since BiPOLES permits neuronal spiking exclusively within the orange-red light window, it facilitates two-color excitation of genetically distinct but spatially intermingled neuronal populations using a second, blue-light-activated ChR (Fig. 4f). To demonstrate this, we expressed somBiPOLES in CA1 VIP interneurons and CheRiff, a blue-light-sensitive ChR ($\lambda_{max} = 460$ nm)[30] in CA1 pyramidal neurons (Fig. 4g, see "Methods" section for details). Both CA1 and VIP neurons innervate Oriens-Lacunosum-Moleculare (OLM) interneurons. Therefore, exclusive excitation of CA1 pyramidal cells or VIP interneurons is expected to trigger excitatory (EPSCs) and inhibitory (IPSCs) postsynaptic currents, respectively. CheRiff-expressing pyramidal cells were readily spiking upon blue, but not orange-red illumination up to 10 mW mm$^{-2}$ (Fig. 4h, Supplementary Fig. 9). Conversely, as expected, red light evoked APs in somBiPOLES-expressing VIP neurons, while blue light up to 100 mW mm$^{-2}$ did not evoke APs (Fig. 4h). Next, we recorded synaptic inputs from these two populations onto VIP-negative GABAergic neurons in stratum-oriens (Fig. 4i). As expected, blue light triggered EPSCs (CheRiff) and red light triggered IPSCs (somBiPOLES), evident by their respective reversal potentials at $8.8 \pm 10.4$ mV and $-71.4 \pm 13.1$ mV (Fig. 4i). Thus, somBiPOLES, in combination with the blue-light sensitive CheRiff enabled independent activation of two distinct populations of neurons in the same field of view.

**Bidirectional neuronal control using dual-laser two-photon holography**. Two-photon holographic excitation enables spatially localized photostimulation of multiple neurons with single-cell resolution in scattering tissue[1]. We evaluated the feasibility of bidirectional control of single neurons by two-photon holographic excitation (Supplementary Fig. 10a) in hippocampal organotypic slices virally transduced with somBiPOLES expressed from a CaMKII promoter. Single-photon excitation confirmed the high potency of somBiPOLES using this expression strategy (Supplementary Fig. 11). The two-photon action spectrum of somBiPOLES was explored by measuring the peak photocurrents ($I_p$) at a range of holding potentials ($-80$ to $-55$ mV) and excitation wavelengths (850 to 1100 nm). Similar to single-photon excitation, blue-shifted wavelengths ($\lambda_{ex} < 980$ nm) generated large photocurrents, apparently dominated by the flow of chloride ions (outward chloride currents below the chloride Nernst potential and inward chloride currents above the chloride Nernst potential, Fig. 5a–c, Supplementary Fig. 10b). Red-shifted wavelengths ($\lambda_{ex} > 980$ nm) generated photocurrents, which appeared to be dominated by the flow of protons and cations across the membrane (inward currents at physiological neuronal membrane potentials, Fig. 5a–c, Supplementary Fig. 10b). Since 920 nm and 1100 nm illumination generated the largest magnitudes of inhibitory and excitatory photocurrents, respectively, these wavelengths were used to evaluate whether the neuronal activity could be reliably suppressed or evoked in neurons expressing

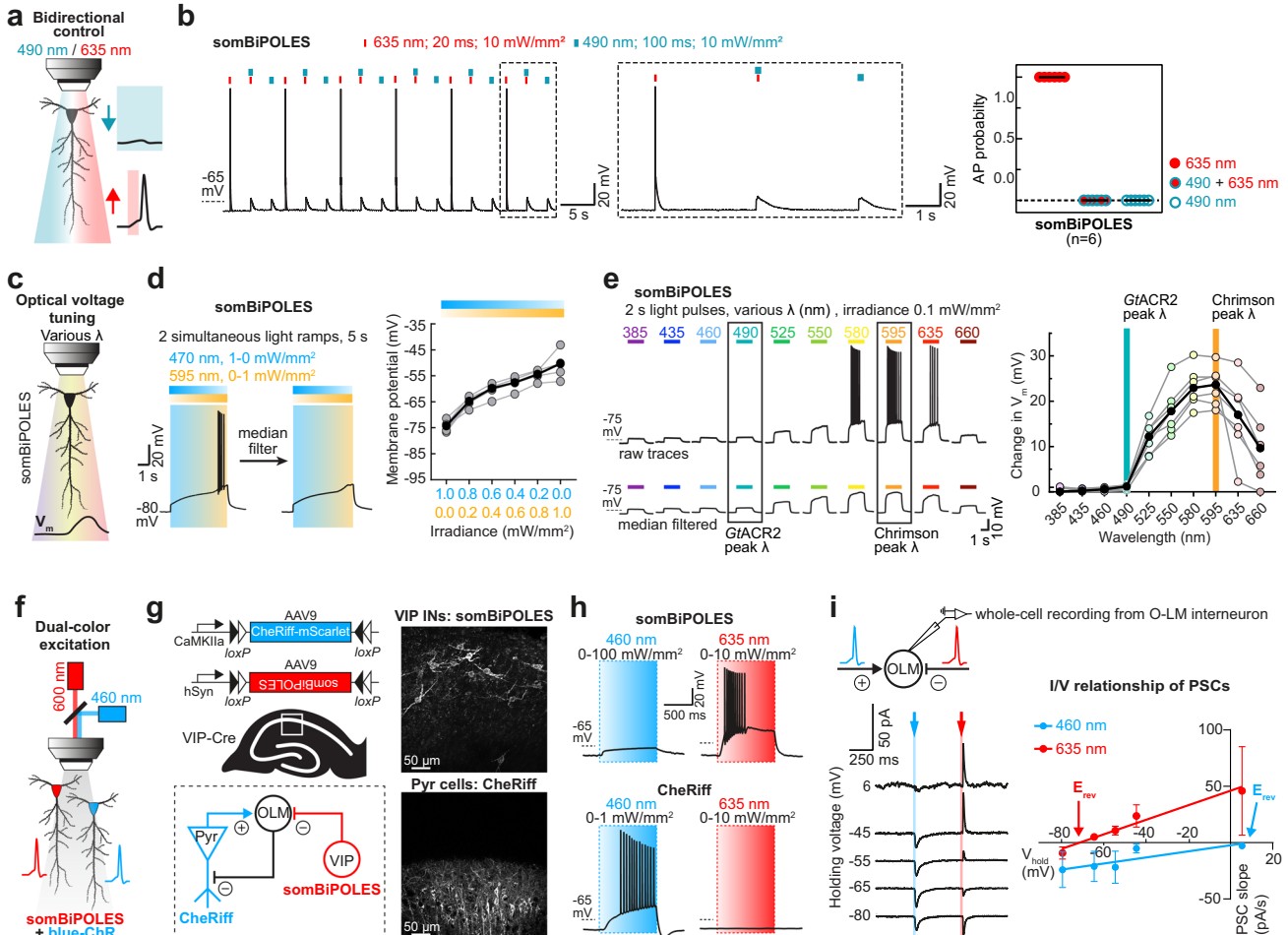

**Fig. 4 Applications of BiPOLES: bidirectional control of neuronal activity, optical voltage tuning, and independent dual-color excitation of two distinct neuronal populations. a** Schematic drawing illustrating bidirectional control of neurons with blue and red light. **b** Current-clamp characterization of bidirectional optical spiking-control with somBiPOLES. Left: Voltage traces showing red-light-evoked APs, which were blocked by a concomitant blue light pulse. Right: quantification of AP probability under indicated conditions (black horizontal lines: medians, $n = 6$ cells). **c** Schematic drawing illustrating control of membrane voltage with somBiPOLES. **d** Left: Representative membrane voltage traces from a somBiPOLES-expressing CA1 pyramidal cell during simultaneous illumination with 470-nm and 595-nm light ramps of the opposite gradient. Voltage traces were median-filtered to reveal the slow change in membrane voltage during the ramp protocol. Right: Quantification of membrane voltage at different 595/470 nm light ratios (black circles: medians, $n = 5$ cells). **e** Left: Representative membrane voltage traces of somBiPOLES in CA1 pyramidal neurons upon illumination with different wavelengths and equal photon flux. As in **d** voltage traces were median-filtered to eliminate action potentials and reveal the slow changes in membrane voltage during the light protocol. Right: Quantification of membrane potential along the spectrum showing optical voltage tuning at the indicated wavelengths. (black circles: medians, an irradiance of 0.1 mW mm$^{-2}$ was kept constant for all wavelengths, $n = 6$ cells). **f** Schematic drawing illustrating control of 2 neurons expressing either somBiPOLES (red) or a blue-light-sensitive ChR (blue). **g** Left: Cre-On/Cre-Off strategy to achieve mutually exclusive expression of CheRiff-mScarlet in CA1 pyramidal neurons and somBiPOLES in VIP-positive GABAergic neurons. Both cell types innervate OLM interneurons in CA1. Right: Example maximum-intensity projection images of two-photon stacks showing expression of somBiPOLES in VIP-interneurons (top) and CheRiff-mScarlet in the pyramidal layer of CA1 (bottom). **h** IC-recordings demonstrating mutually exclusive spiking of somBiPOLES-expressing and CheRiff-expressing neurons under red or blue illumination. **i** Postsynaptic whole-cell voltage-clamp recordings from an OLM cell at indicated membrane voltages showing EPSCs and IPSCs upon blue-light and red-light pulses, respectively. Right: quantification of blue-light and red-light-evoked PSCs and their reversal potential. Symbols show mean ± SEM, $n_{460\ nm} = 8$ cells, $n_{635\ nm} = 7$ cells, lines: linear regression fit, $R^2 = 0.06$ and 0.20 for a blue and red light, respectively.

somBiPOLES. Action potentials could be reliably evoked using short (5 ms) exposure to 1100 nm light (power density: 0.44 mW/μm$^2$), with latency (19.9 ± 6.3 ms) and jitter (2.5 ± 1.5 ms) (Fig. 5d, Supplementary Fig. 10c) comparable to literature values for Chrimson[31]. 5 ms pulses were also able to induce high-fidelity trains of APs with frequencies up to 20 Hz (Supplementary Fig. 10d). It is likely that shorter latency and jitter (and consequently higher rates of trains of APs) could be achieved by replacing the stimulation laser with one with optimized pulse

parameters, in particular, higher peak energy[32]. 920 nm excitation effectively inhibited neural activity, increasing the rheobase of AP firing at power densities above 0.1 mW μm$^{-2}$ (Fig. 5e). It further enabled temporally precise elimination of single electrically evoked APs (Supplementary Fig. 10e) and silencing of neuronal activity over sustained (200 ms) periods (Fig. 5f). Finally, we demonstrate two-photon, bidirectional control of neurons by coincident illumination of appropriately titrated 920 nm and 1100 nm light (Fig. 5g). Thus, somBiPOLES is suitable for dual-color

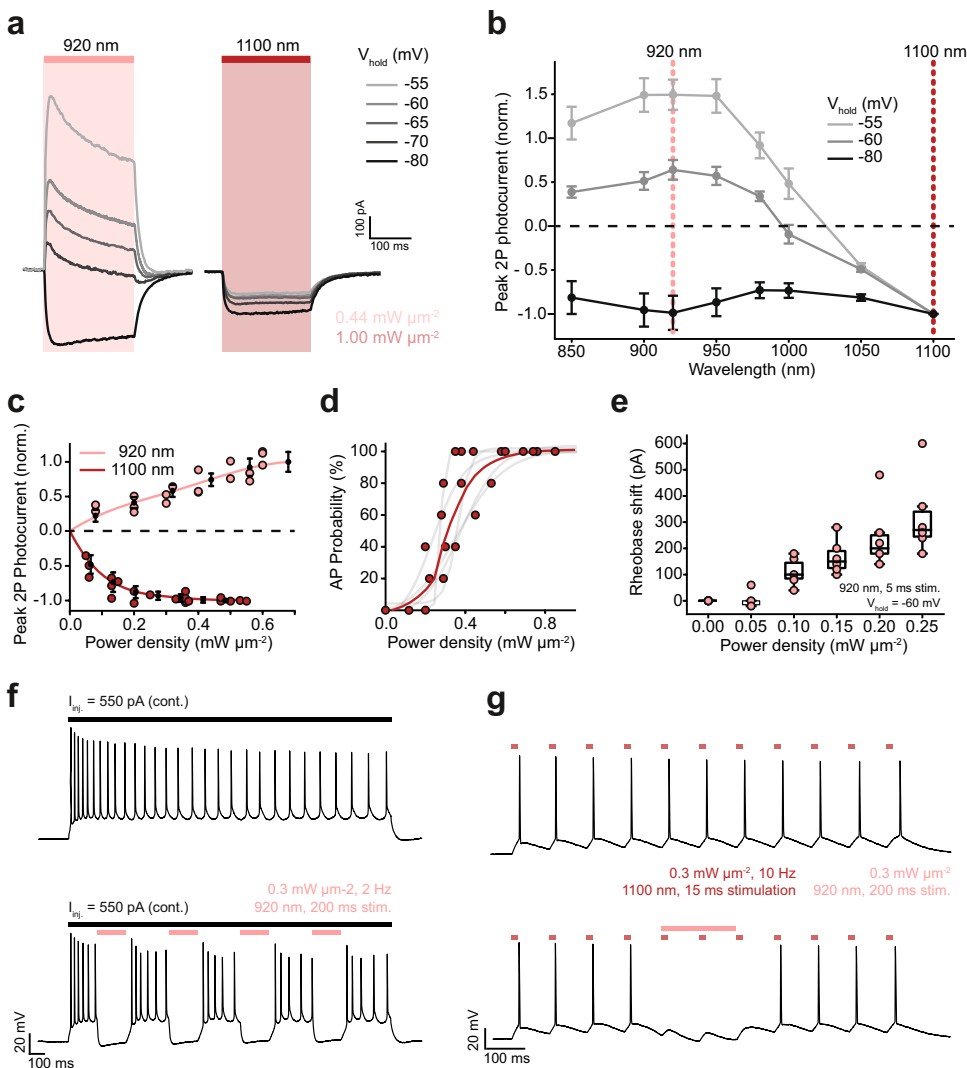

**Fig. 5 Bidirectional control of neuronal activity with somBiPOLES using dual-color two-photon holography. a–c** Voltage clamp (VC) characterization of somBiPOLES in CA1 pyramidal cells. **a** Representative photocurrent traces at different holding potentials, obtained by continuous 200 ms illumination of 920 and 1100 nm at constant average power density (0.44 and 1.00 mW μm$^{-2}$). **b** Peak photocurrent as a function of wavelength at different holding potentials (mean ± SEM, $n = 5$). Data acquired with a constant photon flux of $6.77 \times 10^{26}$ photons s$^1$m$^{-2}$. Dashed lines indicate 920 and 1100 nm respectively; the wavelengths subsequently utilized for photo-stimulation and inhibition. **c** Peak photocurrent as a function of incident power density at a holding potential of −60 mV (mean ± SEM, 920 nm, $n = 4$; 1100 nm, $n = 5$). **d–g** Current clamp (IC) characterization of somBiPOLES in CA1 pyramidal cells. **d** Probability of photo evoked action potentials under 1100 nm illumination for 5 ms ($n = 5$, red: average, gray: individual trials). **e** Characterization of the efficacy of silencing somBiPOLES expressing neurons under 920-nm illumination by co-injection of current (Box: median, 1st–3rd quartile, whiskers: 1.5x inter quartile range, $n = 5$). **f** Representative voltage traces demonstrating sustained neuronal silencing of neurons by two-photon excitation of somBiPOLES at 920 nm. Upper trace (control): 550 pA current injected (illustrated by the black line), no light. Lower trace: continuous injection of 550 pA current, 0.3 mW μm$^{-2}$, 920 nm, 2 Hz, 200 ms illumination. **g** Two-photon, bidirectional, control of single neurons demonstrated by co-incident illumination of 920 nm and 1100 nm light. Upper trace: 10 Hz spike train evoked by 15 ms pulses of 1100 nm light. Lower trace: optically induced action potentials shunted using a single, 200 ms pulse of 920 nm light.

two-photon holographic manipulation of neuronal activity with a cellular resolution with standard lasers typically used for two-photon imaging.

Considering the reliable performance of BiPOLES in pyramidal neurons we next tested its applicability in the invertebrate model systems *C. elegans* and *D. melanogaster*, as well as mice and ferrets, representing vertebrate model systems.

**Bidirectional control of motor activity in *C. elegans*.** We expressed BiPOLES in cholinergic motor neurons of *C. elegans* to optically control body contraction and relaxation. Illumination with

red light resulted in body-wall muscle contraction and effective body shrinkage, consistent with motor neuron activation. Conversely, blue light triggered body extension, indicative of muscle relaxation and thus, cholinergic motor neuron inhibition (Fig. 6b). Maximal body length changes of +3% at 480 nm and −10% at 560–600 nm and reversal of the effect between 480 and 520 nm were consistent with the inhibitory and excitatory action spectrum of BiPOLES (Fig. 6b, Supplementary Fig. 12a). The light effects on body length required functional BiPOLES as the light did not affect body length in the absence of all-*trans*-retinal (ATR, Fig. 6b). Previous strategies for bidirectional motor control in *C. elegans* using ChR2(HR) and NpHR did not show body contraction and elongation in the same

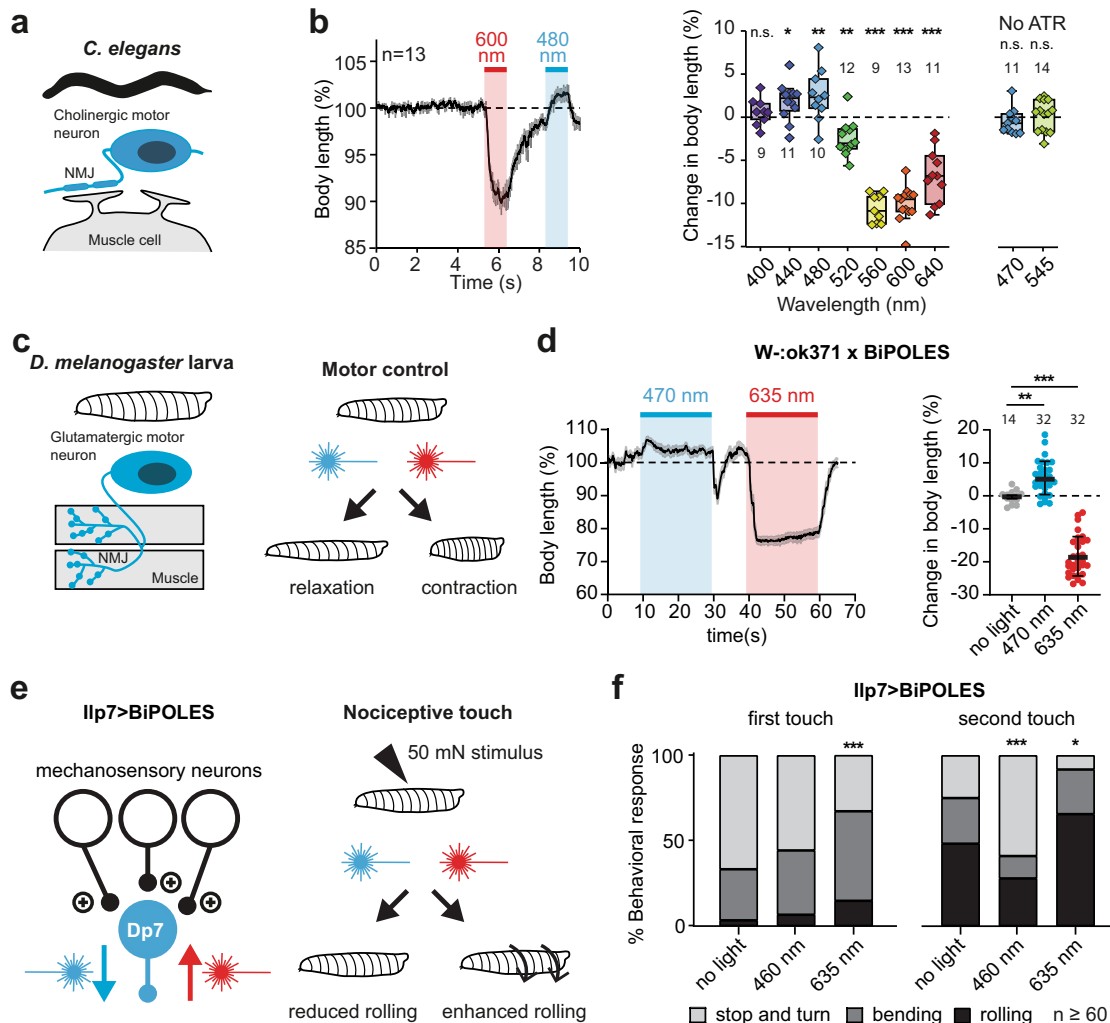

**Fig. 6 BiPOLES allows bidirectional modulation of neuronal activity in *C. elegans* and *D. melanogaster*. a** BiPOLES expressed in cholinergic neurons of *C. elegans* enables bidirectional control of body contraction and relaxation. Scheme of BiPOLES-expressing cholinergic motor neuron innervating a muscle cell. **b** Left: Temporal dynamics of relative changes in body length upon illumination with 600 and 480 nm light (mean ± SEM, 1.1 mW mm$^{-2}$, $n = 13$ animals). Right: Spectral quantification of maximal change in body length, compared is the body length before to during light stimulation (seconds 0–4 vs. seconds 6–9, see Supplementary fig. 12a; Box: median, 1st–3rd quartile, whiskers: 1.5x interquartile range, two-way ANOVA (Sidak's multiple comparisons test), $p$ values: 400 nm ($n = 9$ animals): 0.99, 440 nm ($n = 12$): 0.049, 480 nm ($n = 10$): 0.007, 520 nm ($n = 12$): 0.002, 560 nm ($n = 9$): <0.0001, 600 nm ($n = 13$): <0.0001, 640 nm ($n = 11$): <0.0001, no ATR 470 nm ($n = 11$): 0.24, no ATR 545 nm ($n = 14$): 0.78). Experiments in absence of all-*trans*-retinal were done with 470/40 nm and 545/30 nm bandpass filters. **c** BiPOLES expressed in glutamatergic neurons of *D. melanogaster* larvae enables bidirectional control of body contraction and relaxation. Scheme of BiPOLES-expressing glutamatergic motor neuron innervating muscle fibers. **d** Left: Temporal dynamics of relative changes in body length upon illumination with 470 (17 µW mm$^{-2}$, $n = 32$ animals) and 635 nm light (25 µW mm$^{-2}$, $n = 32$). Right: Quantification of maximal change in body length (mean ± SEM, no light = 14, 470 nm = 32, 635 nm = 32, **$p = 0.0152$, ***$p = 0.0005$, one-way ANOVA with Dunnett's multiple comparisons test). **e** BiPOLES-dependent manipulation of Dp7 neurons in the Drosophila larval brain (Ilp7-Gal4>UAS-BiPOLES) and the resulting change in nociceptive escape behavior following a 50 mN noxious touch. **f** Behavioral response after the first and second mechanical stimulus under blue light (470 nm, 1.7 mW mm$^{-2}$) or red light (635 nm, 2.5 mW mm$^{-2}$) illumination compared to no light. $n = 61$ animals *$p = 0.034$, ***$p = 0.0005$ (first touch) and 0.0007 (second touch), $X^2$-test.

animal[33]. Therefore, we tested this directly with light conditions similar to those used for BiPOLES activation. Excitation with blue light resulted in a 5% body length decrease, while activation of NpHR at its peak wavelength (575 nm) failed to induce significant changes in body length (Supplementary Fig. 12b). Thus, BiPOLES expands the possibilities for bidirectional control of neuronal activity in *C. elegans* beyond what is achievable with currently available tools.

**Bidirectional control of motor activity and nociception in *D. melanogaster*.** Next, we demonstrate bidirectional control of circuit function and behavior with BiPOLES in *Drosophila*

*melanogaster*. *Gt*ACR2 and *Cs*Chrimson were previously used in separate experiments to silence and activate neuronal activity, respectively[10]. In contrast, rhodopsin pump functionality is strongly limited in this organism[10,11], and bidirectional control of neuronal activity has not been achieved. We, therefore, expressed BiPOLES in glutamatergic motor neurons of *D. melanogaster* larvae (Fig. 6c). Illumination with blue light led to muscle relaxation and concomitant elongation (Fig. 6d). The change in body length was similar to animals expressing *Gt*ACR2 alone (Supplementary Fig. 12c). Importantly, *Gt*ACR2 activation in BiPOLES overrides blue-light evoked Chrimson activity and

thereby eliminates blue-light excitation of neurons, as observed with *Cs*Chrimson alone (Supplementary Fig. 12c). Conversely, red illumination of BiPOLES expressing larvae triggered robust muscle contraction and corresponding body length reduction (Fig. 6d). Thus, BiPOLES facilitates bidirectional optogenetic control of neuronal activity in *D. melanogaster* which was not achieved previously.

We further tested BiPOLES functionality in a more sophisticated in vivo paradigm expressing it in key modulatory neurons (dorsal pair Ilp7 neurons, Dp7) of the mechanonociceptive circuit. Dp7 neurons naturally exert bidirectional control of the larval escape response to noxious touch depending on their activation level[34] (Fig. 6e). Acute BiPOLES-dependent silencing of Dp7 neurons with blue light strongly decreased the rolling escape (Fig. 6f), consistent with previously shown chronic silencing of these neurons[34]. In turn, red light illumination of the same animals enhanced escape responses upon noxious touch showing that BiPOLES activation in Dp7 neurons can acutely tune their output and thus the corresponding escape response (Fig. 6f). BiPOLES activation in Dp7 neurons showed a similar ability to block or enhance nociceptive behavior as *Gt*ACR2 or *Cs*Chrimson, respectively, while preventing Chrimson-dependent activation with blue light (Supplementary Fig. 12d, e). Taken together, BiPOLES allows robust, acute, and bidirectional manipulation of neuronal output and behavior in *Drosophila melanogaster* in vivo.

**All-optical, bidirectional control of pupil size in mice**. To further extend the applications of BiPOLES to vertebrates, we generated various conditional and non-conditional viral vectors, in which the expression of the fusion construct is regulated by different promoters (see "Methods" section, Table 1). Using these viral vectors, we sought to test BiPOLES and somBiPOLES in the mammalian brain. To this end, we conditionally expressed somBiPOLES in TH-Cre mice, targeting Cre-expressing neurons in the Locus Coeruleus (LC) (Fig. 7a). Orange illumination (594 nm) through an optical fiber implanted bilaterally above LC reliably triggered transient pupil dilation, indicative of LC-mediated arousal[35] (Fig. 7b–d). Pupil dilation was evident already at 0.7 mW at the fiber tip and gradually increased with increasing light power (Supplementary Fig. 13a). Light-mediated pupil dilation was reverted immediately by additional blue light (473 nm) during the orange-light stimulation or suppressed altogether when blue-light delivery started before orange-light application (Fig. 7b–d), suggesting that orange-light-induced spiking of somBiPOLES-expressing neurons in LC was efficiently shunted. Illumination of the LC in wt-animals did not influence pupil dynamics (Supplementary Fig. 13b). Thus, LC neurons were bidirectionally controlled specifically in somBiPOLES expressing animals.

We estimated the brain volume accessible to reliable activation and inhibition with somBiPOLES using Monte-Carlo simulations of light propagation[16] under the experimental settings used for the LC-manipulations described above (Supplementary Fig. 14). Based on the light parameters required for neuronal excitation and inhibition determined in Fig. 3, and assuming 1 mW of 473 nm and 10 mW of 593 nm at the fiber tip, we estimate that reliable bidirectional control of neuronal activity can be achieved over a distance of >1.5 mm in the axial direction below the fiber tip (Supplementary Fig. 14c).

**Manipulation of neocortical excitation/inhibition ratio in ferrets**. Finally, we applied BiPOLES to bidirectionally control the excitation/inhibition (E/I) ratio in the mammalian neocortex. Therefore, we generated a viral vector using the minimal *Dlx*

promoter[36] (mDlx) to target GABAergic neurons in the ferret secondary visual cortex (V2). Functional characterization in GABAergic neurons in vitro confirms all-optical spiking and inhibition of GABAergic neurons with mDlx-BiPOLES (Supplementary Fig. 15). Thus, we injected mDlx-BiPOLES in ferret V2 to modulate E/I-ratio during sensory processing (Fig. 7e). Extracellular recordings obtained from linear silicon probes in V2 of isoflurane-anesthetized ferrets provided evidence for modulation of cortical activity by shifts in the E/I ratio (Fig. 7f, g). Blue light led to an increase in baseline activity, consistent with the deactivation of inhibitory, GABAergic neurons (Fig. 7f, g). Activation of GABAergic cells by red light did not further decrease the low cortical baseline activity, but significantly reduced cortical responses triggered by sensory stimuli (Fig. 7f, g). Although effects of blue light on evoked spiking were not significant in the average data, we obtained clear evidence in individual recordings that blue light could enhance late response components (Fig. 7f), confirming a disinhibitory effect. Overall, these data suggest that BiPOLES is efficient in bidirectional control of inhibitory mechanisms, demonstrating its applicability for the control of E/I shifts in the cortical microcircuit in vivo.

## Discussion

In summary, BiPOLES is a performance-optimized fusion construct composed of a red-light-activated cation- and a blue-light-activated anion-selective ChR. BiPOLES serves as an optogenetic tool for potent excitation and inhibition of the same neurons with red and blue light, respectively. In addition, it can be applied for exclusive red-light activation of a neuronal subpopulation in multicolor experiments, and for locally defined optical tuning of the membrane voltage between the Nernst potential for chloride and the action potential threshold.

BiPOLES performs reliably in invertebrate and vertebrate model systems, showing potent, bidirectional modulation in the *C. elegans* motor system, the *D. melanogaster* motor and nociceptive systems, and the ferret visual cortex. The addition of the soma-targeting signal from the mammalian potassium channel Kv2.1 yielded somBiPOLES, leading to further enhancement of trafficking to the plasma membrane at the soma and proximal dendrites while avoiding localization to distal dendrites and axons, as previously shown for individually expressed microbial rhodopsins[27–29]. Thus, eliminating the risk of inadvertent blue-light mediated depolarization of axons[28,37] while improving bidirectional optogenetic manipulation of the somatodendritic compartment somBiPOLES is optimized for applications in mammalian systems.

Combining cation and anion channels of overlapping action spectra requires careful consideration of the electrochemical conditions of the neuronal membrane. Since the resting membrane potential is close to the Nernst potential of chloride, anion channels displaying large unitary conductance are needed in order to efficiently shunt depolarizing currents of the red-shifted cation channel, which, in turn, needs to be potent enough to reliably trigger action potentials. Thus, photocurrent amplitudes and spectral sensitivity of the two opsins need to match the aforementioned conditions in order to both reliably silence and drive neuronal activity. If the red-shifted excitatory opsin shows too large, blue-light sensitive photocurrents, it may compromise the silencing capacity of the anion channel. Conversely, if the action spectrum of the blue-light sensitive anion channel extends too far towards longer wavelengths, efficient red-light evoked spiking may get impaired. For the molecular engineering of BiPOLES we focused on a large spectral separation of the anion and the cation conductance. Minimizing the optical cross-talk of both channels favors inhibitory conductance under blue light

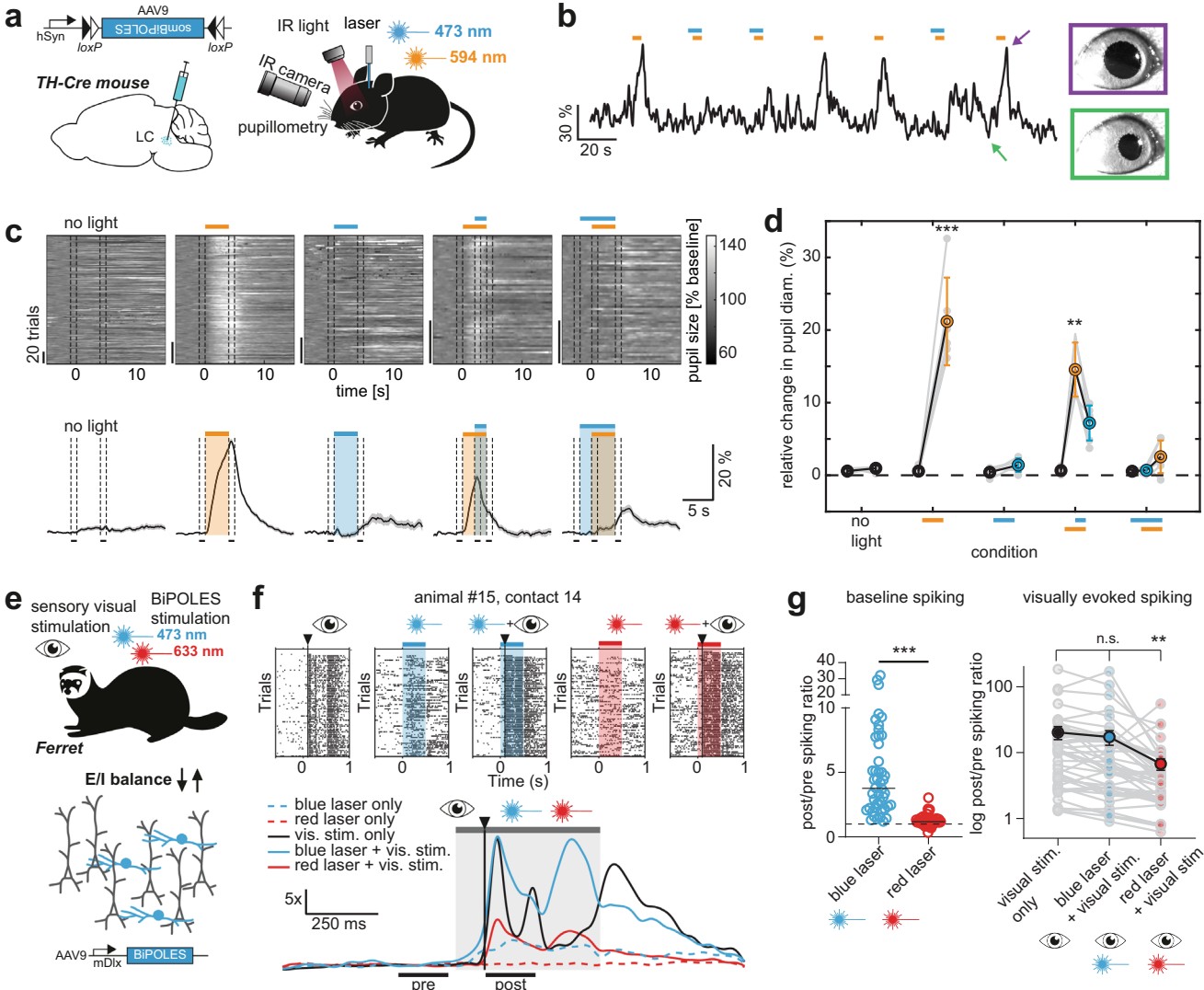

**Fig. 7 BiPOLES and somBiPOLES allow bidirectional modulation of neuronal activity in mice and ferrets. a** Conditional expression of somBiPOLES in Cre-positive neurons of the TH-Cre mouse to modulate pupil dilation. **b** Relative pupil diameter in single trials. Orange and blue bars indicate the time of illumination with 594 (orange) and 473 nm (blue), respectively. Arrows indicate positions of the two example images of the eye. **c** Quantification of normalized pupil size in one animal under various stimulation conditions for somBiPOLES as indicated. Top: single trials. Bottom: mean ± SEM. Dashed lines show time windows used for quantification in the plot on the right. **d** Quantification of relative pupil size ($n = 6$ mice; One-way analysis of variance; $F = 61.67$, $p = 1.36 \times 10^{-12}$; Tukey's multiple comparison test: **$p = 0.0028$, ***$p < 0.0001$). **e** Modulation of GABAergic neurons (blue) in ferret secondary visual cortex (area 18) with mDlx-BiPOLES. Red (633 nm) or blue (473 nm) laser light was used to (de-)activate interneurons with or without a following 10-ms visual flash (white LED; Osram OSLON Compact) to the ferret's right eye. **f** Example neuronal spiking responses at one contact of the linear probe (~700 µm depth) under indicated stimulation conditions Top: Raster-plots of the visual stimulus alone, blue laser (+visual), red laser (+visual) conditions. Bottom: Normalized to 'pre'-phase averaged spike-density plot (sigma = 20 ms) of each indicated condition. Gray area: laser-on epoch; black vertical line: visual stimulus onset. Black horizontal lines indicate the 200 ms pre-stim and post-stim analysis epochs to compute the results in **g**. Note the rate increase after the onset of the blue laser before the onset of the visual stimulus and the reduced answer after red laser illumination. **g** Spike-rate ratio of post vs. pre-laser-stimulus epoch. Left: quantification of laser-mediated impact on baseline spiking rate (no visual stim.). Right: quantification of the spike-rate change of the same units during only visual and laser + visual stimulation ($n = 46$ contacts showing visual responses from 3 animals, **$p = 0.0046$, ***$p < 0.0001$).

illumination and increases both the light intensity range and the spectral range that allows exclusive activation of the red-shifted cation channel. Due to the large spectral separation, BiPOLES can be controlled with two simple light sources, such as LEDs, without the requirement of sophisticated spectral control, making its use straightforward. The *Gt*ACR2-L4-ChRmine-construct might be an interesting alternative if spectrally narrow light sources, such as lasers, are available, because it reaches peak depolarizing currents 60 nm blue-shifted compared to BiPOLES. Thus, inhibition and excitation can be achieved with 430–470 nm

and 530–550 nm (Fig. 1f) providing an additional spectral window in the red, that can be used for a third optogenetic actuator or sensor. Finally, a seemingly trivial but equally important advantage of all the tandem systems we present here is their modular architecture allowing easy tailoring of fusion constructs fulfilling specific future experimental requirements.

Noteworthy, BiPOLES does not represent the first optogenetic tool for bidirectional control of neuronal activity. Different combinations of the excitatory blue-light-sensitive ChR2 and orange-light-sensitive inhibitory ion pumps such as NpHR, bR, or

Arch3.0 were generated previously[2,4,6]. However, among all these variants, only the combination of ChR2 and NpHR (i.e., eNPAC and eNPAC2.0) was successfully used to address neuroscientific questions in mice[6–9]. BiPOLES will significantly expand the possibilities of bidirectional neuronal manipulations, since, aside from efficient expression in a wide array of different model systems, it also features a number of additional advantages: First, combining two potent channels, rather than a pump and a channel, provides a more balanced ionic flux per absorbed photon for the inhibitory and excitatory rhodopsin. This results in a high operational light sensitivity for both excitation and inhibition by orange and blue light, respectively. In contrast, high irradiance and expression levels are required for the ion pumps that only transport one charge per absorbed photon. Second, due to the use of two channels, BiPOLES-mediated photocurrents do not actively move ions against their gradients, which can cause adverse side-effects[37], but rather fixes the neuronal membrane voltage anywhere between the reversal potential of GtACR2 and Chrimson. The membrane voltage can be tuned depending on the ratio of blue/red light or a single light source tuned to wavelengths between the absorption peaks of GtACR2 and Chrimson. Third, inverting the color of the excitatory and inhibitory opsin, compared to previous tools, restricts optical excitation in BiPOLES-expressing cells exclusively to the orange/red spectrum. The inverted color scheme enables scale-free and mutually exclusive spiking of two neuronal populations in combination with a second, blue-light-sensitive ChR, expressed in the second population of neurons, as the blue-light-activated, inhibitory channel GtACR2 potently shunts Chrimson-mediated, blue-light-activated excitatory photocurrents. Other applications could employ multiplexing with blue-light sensitive cyclases[38] or genetically encoded activity-indicators that require blue light for photoconversion[39,40]. Fourth, compared to the first generation of tandem constructs, BiPOLES was optimized for membrane trafficking and especially the somBiPOLES variant shows strongly improved membrane expression in mammalian neurons, enabling reliable and potent optogenetic spiking and inhibition even in deep brain regions in vivo. One additional reason for the superior membrane expression of BiPOLES compared to other rhodopsin-tandems might be the absence of N-terminal, extracellular cysteine residues, which are involved in disulfide bond formation and thus dimerization in all structurally described ChRs[41–44]. The absence of N-terminal cysteines may avoid heteromeric protein networks and undesired clustering of the fused tandem rhodopsins. Fifth, soma-targeted BiPOLES allows efficient and reliable bidirectional control of neuronal spiking over a wide range of light intensities. This is important for in vivo applications in the mammalian brain, where light scattering and absorption lead to an exponential fall-off of the irradiance over distance[17]. The color scheme in somBiPOLES in combination with the large-conductance of GtACR2 and its absence from axon terminals enables potent and reliable silencing with blue light over a wide range of intensities. Potential cross-activation of Chrimson by high blue light intensities did not compromise neuronal silencing in pyramidal neurons. Similarly, due to the red-shifted absorption of Chrimson, neuronal spiking can be efficiently achieved with orange light. somBiPOLES reliably mediates silencing and activation at modest intensities of blue and orange light far away from the fiber tip, while maintaining its wavelength-specificity under high-intensity irradiance, as typically present directly under the fiber tip. Thus, somBiPOLES holds the potential to manipulate neuronal activity in large brain areas with single-photon illumination (Supplementary Fig. 14c). Finally, a fusion protein of two potent channels with opposite charge selectivity targeted to the somatodendritic compartment and displaying a local one-to-one expression ratio in the plasma

membrane enables temporally precise bidirectional control of neuronal activity at single-cell resolution using two-photon excitation. In contrast to widefield illumination with visible light, two-photon excitation in combination with soma-targeted opsins allows optogenetic control with single-cell resolution[45–47]. Bidirectional optogenetic control in the same cells has not been achieved with two-photon excitation, so far; partially due to the low quantum efficiency of rhodopsin pumps, which limits their two-photon activation. In contrast, the large conductance of the two channels improves their efficacy with respect to the number of transported ions per absorbed photon, and their presence at equal stoichiometry anywhere on the membrane ensures the reliable and reproducible generation of anion currents and/or cation currents, which is particularly important under locally confined two-photon excitation.

In principle, also multicistronic vectors encoding both opsins under a single promoter using either an internal ribosomal entry site (IRES)[48] or a 2A ribosomal skip sequence allow expression of both ion channels at a fixed ratio from a single AAV vector[3,7]. However, with both of these strategies, neither co-localized nor stoichiometric membrane expression of both channels is guaranteed since both channels might get differentially targeted and distributed in the plasma membrane. This may not pose a limitation for experiments that require bidirectional control of large numbers of cells where precise control of a single-cell activity or sub-cellular ion gradients is not so crucial. BiPOLES as a covalently linked fusion protein displays a fixed expression of both opsins at a 1:1 stoichiometry anywhere in the membrane and membrane trafficking or degradation of both opsins occur at identical rates, preserving excitatory and inhibitory currents at a fixed ratio in all expressing cells. A fixed stoichiometry anywhere in the cell membrane is important if local, subcellular activation of the opsins is required, such as during two-photon excitation or when a fixed ratio of cation and anion conductance is desired between different neurons or in particular neuronal compartments, such as single dendrites or dendritic spines.

Notably, BiPOLES employs an anion channel for optogenetic silencing and therefore relies on the extracellular and intracellular chloride concentration. In the case of a depolarized chloride Nernst potential, the opening of the anion channel may produce depolarizing currents, which can trigger action potentials or neurotransmitter release[49]. Unlike for rhodopsin pumps, efficient silencing consequently requires low cytosolic chloride concentrations and is therefore limited in neurons or cellular compartments with a depolarized Nernst potential for chloride, such as immature neurons or axon terminals. Given these caveats, BiPOLES may not be suitable for bidirectional control of developing neurons or presynaptic boutons. In this case, silencing may be more efficient with rhodopsin pumps, despite their own limitations[37,49] or with G-protein coupled rhodopsins[50,51]. As with any optogenetic application, neurophysiological parameters need to be considered by the experimenter, guiding the appropriate choice of the tool suitable to address the specific experimental requirements.

Since BiPOLES can be used to spike or inhibit the same population of mature neurons in vivo, a number of previously inaccessible questions can be addressed. During extracellular recordings, BiPOLES may be useful for optogenetic identification (optotagging) with red light[52] and optogenetic silencing of the same neurons. This will permit verification of the identity of silenced neurons by their spiking profiles. Moreover, in combination with a second, blue-light sensitive ChR, BiPOLES can be used to map local networks of spatially intermingled neurons. For example, expressed in distinct types of molecularly defined GABAergic neurons, connectivity of these neurons to a postsynaptic target cell can be evaluated. Additional applications of

BiPOLES may encompass bidirectional control of engram neurons[53] to test both necessity and sufficiency of a particular set of neurons for memory retrieval or switching the valence of a particular experience by inhibiting or activating the same or even two distinct populations of neuromodulatory neurons. In principle, this could even be achieved with cellular resolution using two-photon holography. Due to its utility for a wide range of research questions, its versatile functionality, and its applicability in numerous model systems, as demonstrated in this study, BiPOLES fills an important gap in the optogenetic toolbox and might become the tool of choice to address a number of yet inaccessible problems in neuroscience.

## Methods

**Molecular biology.** For HEK-cell expression, the coding sequences of Chrimson (KF992060.1), CsChrimson (KJ995863.2) from *Chlamydomonas noctigama*[12], ChRmine from *Rhodomonas lens* although initially attributed to *Tiarina fusus*[22,25] (Addgene #130997), bReaChES[20], iC++ (Addgene #98165)[19], Aurora (Addgene #98217)[11], GtACR1 (KP171708) and GtACR2 (KP171709) from *Guillardia theta*[18], as well as the blue-shifted Arch3.0 mutant M128A/S151A/A225T herein described as ArchBlue[26] were cloned together with mCerulean3[54] and a trafficking signal (ts) from the Kir 2.1 channel[4] into a pCDNA3.1 vector containing the original opsin tandem cassette[2] with a linker composed of eYFP and the first 105 N-terminal amino acids of the rat gastric H+/K+-ATPase beta subunit (βHK, NM_012510.2), kindly provided by Sonja Kleinlogel (University of Bern, CH). For direct comparison also the bicistronic tool eNPAC2.0[6]—kindly provided by Karl Deisseroth (Stanford University, CA)—was cloned into the same backbone. Site-directed mutagenesis to introduce the f-Chrimson and vf-Chrimson mutations Y261F, S267M, and K176R[21] was performed using the QuickChange Site-Directed Mutagenesis Kit (Agilent Technologies, Santa Clara, CA) according to the manufacturers' instructions.

For neuronal expression, the insert consisting of GtACR2-ts-mCerulean3-βHK-Chrimson was cloned into an AAV2-backbone behind human synapsin (hSyn) promoter (pAAV-hSyn-BiPOLES-mCerulean; Addgene #154944). A soma-targeted, membrane-trafficking optimized variant was generated by fusing an additional trafficking signal from the potassium channel Kv2.1[27] to the C-terminus of Chrimson (pAAV-hSyn-somBiPOLES-mCerulean; Addgene #154945). For expression in GABAergic neurons, BiPOLES was cloned into an AAV2-backbone behind the minimal Dlx (mDlx) promoter[36] resulting in pAAV-mDlx-BiPOLES-mCerulean (Addgene #154946). For expression in projection neurons, somBiPOLES was cloned into an AAV2-backbone behind the minimal CaMKII promoter[55] resulting in pAAV-CaMKII-somBiPOLES-mCerulean (Addgene #154948). Double-floxed inverted open reading frame variants of BiPOLES and somBiPOLES were generated by cloning these inserts in antisense direction behind the hSyn promoter, flanked by two loxP and lox2272 sites (hSyn-DIO-BiPOLES-mCerulean, Addgene #154950; hSyn-DIO-somBiPOLES-mCerulean, Addgene #154951). Note that in all constructs the mCerulean3-tag is fused between GtACR2-ts and βHK-Chrimson and therefore part of BiPOLES. We nonetheless chose to add "mCerulean" to the plasmid names to remind the reader of the presence of a cyan fluorophore in BiPOLES. BiPOLES stands for "Bidirectional Pair of Opsins for Light-induced Excitation and Silencing". Sequences of all primers used for cloning and sequences of DNA inserts used in this study are provided in a separate list (Supplementary Data 1).

**Patch-Clamp experiments in HEK293 cells.** Fusion constructs were expressed under the control of a CMV-promotor in HEK293 cells that were cultured in Dulbecco's Modified Medium (DMEM) with stable glutamine (Biochrom, Berlin, Germany), supplemented with 10% (v/v) fetal bovine serum (FBS Superior; Biochrom, Berlin, Germany), 1 µM all-*trans*-retinal, and 100 µg ml⁻¹ penicillin/

streptomycin (Biochrom, Berlin, Germany). Cells were seeded on poly-lysine coated glass coverslips at a concentration of $1 \times 10^5$ cell ml⁻¹ and transiently transfected using the FuGENE® HD Transfection Reagent (Promega, Madison, WI). two days before measurement.

Patch-clamp experiments were performed in transgene expressing HEK293 cells two days after transfection[56]. Patch pipettes were prepared from borosilicate glass capillaries (G150F-3; Warner Instruments, Hamden, CT) using a P-1000 micropipette puller (Sutter Instruments, Novato, CA) and subsequently fire-polished. Pipette resistance was between 1.2 and 2.5 MΩ. Single fluorescent cells were identified using an Axiovert 100 inverted microscope (Carl Zeiss, Jena, Germany). Monochromatic light (± 7 nm) was provided by a Polychrome V monochromator (TILL Photonics, Planegg, Germany) or by a pE-4000 CoolLED system (CoolLED, Andover, UK) for light titration experiments. Light intensities were attenuated by a motorized neutral density filter wheel (Newport, Irvine, CA) for equal photon flux during action spectra recordings. Light pulses of the Polychrome V were controlled by a VS25 and VCM-D1 shutter system (Vincent Associates, Rochester, NY). Recordings were done with an AxoPatch 200B amplifier (Molecular Devices, Sunnyvale, CA) or an ELV-03XS amplifier (npi Electronics, Tamm, Germany), filtered at 2 kHz, and digitized using a DigiData 1440 A digitizer (Molecular Devices, Sunnyvale, CA) at a sampling rate of 10 kHz. The reference bath electrode was connected to the bath solution via a 140 mM NaCl agar bridge. Bath solutions contained 140 mM NaCl, 1 mM KCl, 1 mM CsCl, 2 mM CaCl₂, 2 mM MgCl₂ and 10 mM HEPES at pH_e 7.2 (with glucose added up to 310 mOsm). Pipette solution contained 110 mM NaGluconate, 1 mM KCl, 1 mM CsCl, 2 mM CaCl₂, 2 mM MgCl₂, 10 mM EGTA and 10 mM HEPES at pH_i 7.2 (glucose added up to 290 mOsm). All light intensities were measured in the object plane using a P9710 optometer (Gigahertz-Optik, Türkenfeld, Germany) and normalized to the water Plan-Apochromat ×40/1.0 differential interference contrast (DIC) objective illuminated field (0.066 mm²). The irradiance was 2.7 mW mm⁻² at 650 nm, 3.5 mW mm⁻² at 600 nm, 4.2 mW mm⁻² at 530 nm, 5.7 mW mm⁻² at 490 nm, and 5.2 mW mm⁻² at 450 nm. All electrical recordings were controlled by the pCLAMP™ software (Molecular Devices, Sunnyvale, CA). All whole-cell recordings had a membrane resistance of at least 500 MΩ (usual >1 GΩ) and an access resistance below 10 MΩ.

**Preparation of organotypic hippocampal slice cultures.** All procedures were in agreement with the German national animal care guidelines and approved by the independent Hamburg state authority for animal welfare (Behörde für Justiz und Verbraucherschutz). They were performed in accordance with the guidelines of the German Animal Protection Law and the animal welfare officer of the University Medical Center Hamburg-Eppendorf.

Organotypic hippocampal slices were prepared from Wistar rats or VIP-IRES-Cre mice of both sexes (Jackson-No. 031628) at post-natal days 5–7[57]. Dissected hippocampi were cut into 350 µm slices with a tissue chopper and placed on a porous membrane (Millicell CM, Millipore). Cultures were maintained at 37 °C, 5% CO₂ in a medium containing 80% MEM (Sigma M7278), 20% heat-inactivated horse serum (Sigma H1138) supplemented with 1 mM L-glutamine, 0.00125% ascorbic acid, 0.01 mg ml⁻¹ insulin, 1.44 mM CaCl₂, 2 mM MgSO₄ and 13 mM D-glucose. No antibiotics were added to the culture medium.

**Transgene delivery for single-photon experiments.** For transgene delivery in organotypic slices, individual CA1 pyramidal cells were transfected by single-cell electroporation[58] between DIV 14–16. Except for pAAV-hSyn-eNPAC2.0, which was used at a final concentration of 20 ng µl⁻¹, all other plasmids, namely pAAV-hSyn-BiPOLES-mCerulean, pAAV-hSyn-somBiPOLES-mCerulean, pAAV-hSyn-Chrimson-mCerulean, and pAAV-hSyn-somGtACR2-mCerulean were used at a final concentration of 5 ng µl⁻¹ in K-gluconate-based solution consisting of (in mM): 135 K-gluconate, 10 HEPES, 4 Na₂-ATP, 0.4 Na-GTP, 4 MgCl₂, 3 ascorbate, 10 Na₂-phosphocreatine (pH 7.2). A plasmid encoding hSyn-mKate2 or hSyn-mCerulean (both at 50 ng µl⁻¹) was co-electroporated with the opsin-mCerulean or eNPAC2.0 plasmids, respectively, and served as a morphology marker. An Axoporator 800 A (Molecular Devices) was used to deliver 50 hyperpolarizing

**Table 1 List of recombinant adeno-associated viral vectors used for experiments in organotypic hippocampal slices.**

| Recombinant adeno-associated virus (rAAV2/9) | Titer used for transduction of hippocampal organotypic slice cultures (vg/ml) | Addgene plasmid reference |
|---|---|---|
| mDlx-BiPOLES-mCerulean | $2.8 \times 10^{13}$ | 154946 |
| hSyn-DIO-BiPOLES-mCerulean | $7.0 \times 10^{13}$ | 154950 |
| hSyn-DIO-somBiPOLES-mCerulean | $3.4 \times 10^{13}$ | 154951 |
| CaMKIIa(0.4)-somBiPOLES-mCerulean | $2.5 \times 10^{13}$ | 154948 |
| CaMKIIa(0.4)-DO-CheRiff-ts-mScarlet-ER | $8.15 \times 10^{11}$ | n.a. |
| mDlx-H2B-EGFP | $2.8 \times 10^{10}$ | n.a. |
| CaMKIIa-Cre | $3.0 \times 10^{12}$ | n.a. |

Viruses were transduced at the indicated titers. *n.a.*: not applicable.

pulses (−12 V, 0.5 ms) at 50 Hz. During electroporation, slices were maintained in pre-warmed (37 °C) HEPES-buffered solution (in mM): 145 NaCl, 10 HEPES, 25 D-glucose, 2.5 KCl, 1 MgCl$_2$, and 2 CaCl$_2$ (pH 7.4, sterile filtered). In some cases, slice cultures were transduced with recombinant adeno-associated virus (see Table 1 for details) at DIV 3–5[59]. The different rAAVs were locally injected into the CA1 region using a Picospritzer (Parker, Hannafin) by a pressurized air pulse (2 bar, 100 ms) expelling the viral suspension into the slice. During virus transduction, membranes carrying the slices were kept on pre-warmed HEPES-buffered solution.

**Preparation of organotypic hippocampal slice cultures for two-photon holographic stimulation of somBiPOLES.** All experimental procedures were conducted in accordance with guidelines from the European Union and institutional guidelines on the care and use of laboratory animals (council directive 2010/63/EU of the European Union). Organotypic hippocampal slices were prepared from mice (Janvier Labs, C57Bl6J) at postnatal day 8 (P8). Hippocampi were sliced into 300 μm thick sections in a cold dissecting medium consisting of GBSS supplemented with 25 mM D-glucose, 10 mM HEPES, 1 mM Na-Pyruvate, 0.5 mM α-tocopherol, 20 nM ascorbic acid, and 0.4% penicillin/streptomycin (5000 U ml$^{-1}$).

Slices were placed onto a porous membrane (Millicell CM, Millipore) and cultured at 37 °C, 5% CO2 in a medium consisting of 50% Opti-MEM (Fisher 15392402), 25% heat-inactivated horse serum (Fisher 10368902), 24% HBSS, and 1% penicillin/streptomycin (5000 U ml$^{-1}$). This medium was supplemented with 25 mM D-glucose, 1 mM Na-Pyruvate, 20 nM ascorbic acid, and 0.5 mM α-tocopherol. After three days in-vitro, the medium was replaced with one containing 82% neurobasal-A, 15% heat-inactivated horse serum (Fisher 11570426), 2% B27 supplement (Fisher, 11530536), 1% penicillin/streptomycin (5000 U ml$^{-1}$), which was supplemented with 0.8 mM L-glutamine, 0.8 mM Na-Pyruvate, 10 nM ascorbic acid and 0.5 mM α-tocopherol. This medium was removed and replaced once every 2-3 days.

Slices were transduced with rAAV9-CaMKII-somBiPOLES-mCerulean at DIV 3 by bulk application of 1 μl of virus (final titer: $2.5 \times 10^{13}$ vg ml$^{-1}$) per slice. Experiments were performed between DIV 13 and 17.

**Slice culture electrophysiology with singe-photon stimulation.** At DIV 19-21, whole-cell patch-clamp recordings of transfected or virus-transduced CA1 pyramidal or GABAergic neurons were performed. Experiments were done at room temperature (21–23 °C) under visual guidance using a BX 51WI microscope (Olympus) equipped with Dodt-gradient contrast and a Double IPA integrated patch amplifier controlled with SutterPatch software (Sutter Instrument, Novato, CA). Patch pipettes with a tip resistance of 3–4 MΩ were filled with an intracellular solution consisting of (in mM): 135 K-gluconate, 4 MgCl$_2$, 4 Na$_2$-ATP, 0.4 Na-GTP, 10 Na$_2$-phosphocreatine, 3 ascorbate, 0.2 EGTA, and 10 HEPES (pH 7.2). Artificial cerebrospinal fluid (ACSF) consisted of (in mM): 135 NaCl, 2.5 KCl, 2 CaCl$_2$, 1 MgCl$_2$, 10 Na-HEPES, 12.5 D-glucose, 1.25 NaH$_2$PO$_4$ (pH 7.4). In experiments where synaptic transmission was blocked, 10 μM CPPene, 10 μM NBQX, and 100 μM picrotoxin (Tocris, Bristol, UK) were added to the recording solution. In experiments analyzing synaptic inputs onto O-LM interneurons, ACSF containing 4 mM CaCl$_2$ and 4 mM MgCl$_2$ was used to reduce the overall excitability. Measurements were corrected for a liquid junction potential of −14,5 mV. Access resistance of the recorded neurons was continuously monitored and recordings above 30 MΩ were discarded. A 16 channel LED light engine (CoolLED pE-4000, Andover, UK) was used for epifluorescence excitation and delivery of light pulses for optogenetic stimulation (ranging from 385 to 635 nm). Irradiance was measured in the object plane with a 1918 R power meter equipped with a calibrated 818 ST2 UV/D detector (Newport, Irvine CA) and divided by the illuminated field of the Olympus LUMPLFLN 60XW objective (0.134 mm$^2$).

For photocurrent density measurements in voltage-clamp mode CA1 cells expressing BiPOLES, somBiPOLES, Chrimson or som$Gt$ACR2 were held at −75 or −55 mV to detect inward (cationic) or outward (anionic) currents elicited by red (635 nm, 20 ms, 1 and 10 mW mm$^{-2}$) and blue light (490 nm, 100 ms, 10 mW mm$^{-2}$), respectively. For each cell, the peak photocurrent amplitude (in pA) was divided by the cell membrane capacitance (in pF) which was automatically recorded by the SutterPatch software in voltage-clamp mode ($V_{hold} = -75$ mV).

In current-clamp experiments holding current was injected to maintain CA1 cells near their resting membrane potential (−75 to −80 mV). To assess the suitability of BiPOLES and somBiPOLES as dual-color neuronal excitation and silencing tools, alternating pulses of red (635 nm, 20 ms, 10 mW mm$^{-2}$), blue (490 nm, 100 ms, 10 mW mm$^{-2}$), and a combination of these two (onset of blue light 40 ms before red light) were delivered to elicit and block action potentials. For eNPAC2.0 alternating pulses of blue (470 nm, 20 ms, 10 mW mm$^{-2}$), yellow (580 nm, 100 ms, 10 mW mm$^{-2}$), and a combination of these two (onset of yellow light 40 ms before blue light) were used.

In experiments determining the spiking probability of somBiPOLES and Chrimson under illumination with light of different wavelengths (470, 595, and 635 nm), a train of 20 light pulses (5 ms pulse duration) was delivered at 5 Hz. For each wavelength, irradiance values from 0.1 to 100 mW mm$^{-2}$ were used. For comparisons with eNPAC2.0, only light of 470 nm was used, which is the peak activation wavelength of ChR2(HR). AP probability was calculated by dividing the number of light-triggered APs by the total number of light pulses.

To compare the irradiance threshold needed to spike CA1 cells with BiPOLES, somBiPOLES, eNPAC2.0, Chrimson, and CheRiff across different wavelengths, 470, 525, 595, and 635 nm light ramps going from 0 to 10 mW mm$^{-2}$ over 1 s were delivered in current-clamp mode. In the case of BiPOLES and somBiPOLES the blue light ramp went up to 100 mW mm$^{-2}$ to rule out that very high blue-light irradiance might still spike neurons. The irradiance value at the time of the first spike was defined as the irradiance threshold (in mW mm$^{-2}$) needed to evoke action potential firing.

To measure the ability of BiPOLES, somBiPOLES, and som$Gt$ACR2 to shift the rheobase upon blue-light illumination, depolarizing current ramps (from 0–100 to 0–900 pA) were injected into CA1 neurons in the dark and during illumination with 490 nm light at irradiance values ranging from 0.001 to 100 mW mm$^{-2}$. The injected current at the time of the first spike was defined as the rheobase. The relative change in the number of ramp-evoked APs was calculated counting the total number of APs elicited during the 9 current ramp injections (from 0–100 to 0–900 pA) for each irradiance and normalized to the number of APs elicited in the absence of light. The same experiment was conducted for eNPAC2.0, but using 580 nm light ranging from 0.01 to 100 mW mm$^{-2}$. Statistical significance was calculated using the Friedman test.

To optically clamp the neuronal membrane potential using somBiPOLES, simultaneous illumination with blue and orange light at varying ratios was used. In current-clamp experiments, 470 and 595 nm light ramps of opposite gradients (1 to 0 mW mm$^{-2}$ and 0 to 1 mW mm$^{-2}$, respectively) were applied. Alternatively, optical clamping of the membrane potential was achieved by tuning a single wavelength between 385 and 660 nm (2 s light pulses, 0.1 mW mm$^{-2}$). Voltage traces were median-filtered to remove orange/red-light-mediated spikes and reveal the slow change in membrane voltage during illumination.

For independent optogenetic activation of two distinct populations of neurons, organotypic slice cultures from VIP-Cre mice were transduced with two rAAVs: 1, a double-floxed inverted open reading frame (DIO) construct encoding somBiPOLES (hSyn-DIO-somBiPOLES-mCerulean, see Table 1 for details) to target VIP-positive interneurons, and 2, a double-floxed open reading frame (DO) construct encoding CheRiff (hSyn-DO-CheRiff-ts-mScarlet-ER, see Table 1 for details) to target CA1 pyramidal neurons and exclude expression in VIP-positive cells. Synaptic input from these two populations was recorded in VIP-negative stratum-oriens GABAergic neurons (putative O-LM cells). In CA1, O-LM neurons receive innervation both from local CA1 pyramidal neurons and VIP-positive GABAergic neurons[60]. To facilitate the identification of putative GABAergic postsynaptic neurons in stratum oriens, slices were transduced with an additional rAAV encoding mDlx-H2B-EGFP. In the absence of synaptic blockers light-evoked EPSCs and IPSCs were recorded while holding the postsynaptic cell at different membrane potentials (−80, −65, −55, −45, and 6 mV) in whole-cell voltage-clamp mode. A blue (460 nm, 0.03–84.0 mW mm$^{-2}$) and a red (635 nm, 6.0–97.0 mW mm$^{-2}$) light pulse were delivered 500 ms apart from each other through a Leica HC FLUOTAR L ×25/0.95 W VISIR objective.

To functionally assess the putative expression of somBiPOLES in the axon terminals of CA3 pyramidal cells, slice cultures were transduced with an rAAV9 encoding for CaMKIIa(0.4)-somBiPOLES-mCerulean (see Table 1 for details). Red-light evoked EPSCs were recorded in postsynaptic CA1 cells during local illumination either in CA3 at the somata (two light pulses of 5 ms delivered 40 ms apart using a fiber-coupled LED (400 μm fiber, 0.39 NA, 625 nm, Thorlabs) controlled by a Mightex Universal 4-Channel LED Driver (1.6 mW at fiber tip), or in CA1 at axon terminals of somBiPOLES-expressing CA3 cells (two light pulses of 5 ms delivered 40 ms apart through the ×60 microscope objective, 635 nm, 50 mW mm$^{-2}$). Axonal light stimulation was done in the presence of tetrodotoxin (TTX, 1 μM) and 4-aminopyridine (4-AP, 100 μM) to avoid antidromic spiking of CA3 cells.

To determine the high-frequency spiking limit with somBiPOLES, action potentials were triggered in CA1 cells at frequencies ranging from 10 to 100 Hz using 40 light pulses (595 nm, 3 ms pulse width, 10 mW mm$^{-2}$). AP probability was calculated by dividing the number of light-triggered APs by the total number of light pulses.

To characterize the spectral activation of BiPOLES, eNPAC2.0. and som$Gt$ACR2, photocurrents were recorded from CA1 cells in a voltage-clamp mode in response to 500 ms illumination with various wavelengths (from 385 to 660 nm, 10 mW mm$^{-2}$). BiPOLES-expressing and som$Gt$ACR2-expressing cells were held at a membrane voltage of −55 mV, more positive than the chloride Nernst potential, to measure light-mediated outward chloride currents. Photocurrent recordings from eNPAC2.0-expressing cells were done at a holding voltage of −75 mV. For BiPOLES and eNPAC2.0 the photocurrent ratio between excitatory and inhibitory photocurrents was calculated in each cell by diving the amplitude of the photocurrents evoked by 490/595 nm (for BiPOLES) and 460/580 nm (for eNPAC2.0).

Passive and active membrane parameters were measured in somBiPOLES-expressing and non-transduced, wild-type CA1 pyramidal cells. Resting membrane potential, membrane resistance, and capacitance were automatically recorded by the SutterPatch software in voltage-clamp mode ($V_{hold} = -75$ mV) in response to a voltage test pulse of 100 ms and −5 mV. The number of elicited action potentials were counted in response to a somatic current injection of 300 pA in current-clamp mode (0 pA holding current). For the 1st elicited AP, the voltage threshold, peak, and amplitude were measured.

**Slice culture immunohistochemistry and confocal imaging**. The subcellular localization of BiPOLES and somBiPOLES in hippocampal neurons was assessed 20 days after virus transduction (rAAV9-hSyn-DIO-BiPOLES-mCerulean + CaMKIIa-Cre, and CaMKIIa(0.4)-somBiPOLES-mCerulean, respectively. See Table 1 for details). Hippocampal organotypic slice cultures were fixed in a solution of 4% (w/v) paraformaldehyde (PFA) in PBS for 30 min at room temperature (RT). Next, slices were washed in PBS (3 × 10 min), blocked for 2 h at RT (10% [v/v] normal goat serum [NGS] in 0.3% [v/v] Triton X-100 containing PBS) and subsequently incubated for 48 h at 4 °C with a primary antibody against GFP to amplify the mCerulean signal (chicken, anti-GFP, Invitrogen, A10262, Lot 1972783) at 1:1000 in carrier solution (2% [v/v] NGS, in 0.3% [v/v] Triton X-100 containing PBS). Following 3 rinses of 10 min with PBS, slices were incubated for 3 h at RT in carrier solution (same as above) with an Alexa Fluor® dye-conjugated secondary antibody (goat, anti-chicken Alexa-488, Invitrogen; A11039, Lot 2079383, 1:1000). Slices were washed again, transferred onto glass slides, and mounted for visualization with Shandon Immu-Mount (Thermo Scientific; 9990402).

Confocal images were acquired using a laser-scanning microscope (Zeiss, LSM 900) equipped with a ×40 oil-immersion objective lens (Zeiss EC Plan-Neofluar ×40/1.3 oil). Excitation/emission filters were appropriately selected for Alexa 488 using the dye selection function of the ZEN software. The image acquisition settings were optimized once and kept constant for all images within an experimental data set. Z-stack images were obtained using a 1 μm z-step at a 1024 × 1024-pixel resolution scanning at 8 μs per pixel. Fiji[61] was used to quantify fluorescence intensity values along a line perpendicular to the cell equator and spanning the cell diameter. For each cell, gray values above 80% of the maximum intensity were distributed in 10 bins according to their location along the line.

**Slice culture two-photon imaging**. Neurons in organotypic slice cultures (DIV 19-21) were imaged with two-photon microscopy to check for the live expression of hSyn-DIO-somBiPOLES-mCerulean, CaMKIIa(0.4)-DO-CheRiff-ts-mScarlet-ER, mDlx-BiPOLES-mCerulean and CaMKIIa(0.4)-somBiPOLES-mCerulean. The custom-built two-photon imaging setup was based on an Olympus BX-51WI upright microscope upgraded with a multiphoton imaging package (DF-Scope, Sutter Instrument), and controlled by ScanImage 2017b software (Vidrio Technologies). Fluorescence was detected through the objective (Leica HC FLUOTAR L 25x/0.95 W VISIR) and through the oil immersion condenser (numerical aperture 1.4, Olympus) by two pairs of GaAsP photomultiplier tubes (Hamamatsu, H11706-40). Dichroic mirrors (560 DXCR, Chroma Technology) and emission filters (ET525/70m-2P, ET605/70m-2P, Chroma Technology) were used to separate cyan and red fluorescence. Excitation light was blocked by short-pass filters (ET700SP-2P, Chroma Technology). A tunable Ti:Sapphire laser (Chameleon Vision-S, Coherent) was set to 810 nm to excite mCerulean on BiPOLES and somBiPOLES. An Ytterbium-doped 1070-nm pulsed fiber laser (Fidelity-2, Coherent) was used at 1070 nm to excite mScarlet on CheRiff. Maximal intensity projections of z-stacks were generated with Fiji[61].

**Electrophysiology for two-photon photostimulation of somBiPOLES**. At DIV 13–17, whole-cell patch-clamp recordings of somBiPOLES-infected excitatory neurons were performed at room temperature (21– 23 °C). An upright microscope (Scientifica, SliceScope) was equipped with an infrared (IR) source (Thorlabs, M1050L4), oblique condenser, microscope objective (Nikon, CFI APO NIR, ×40, 0.8 NA), tube lens (Thorlabs, AC508-300-B), and a CMOS camera (Point Grey, CM3-U3-31S4M-CS) to collect IR light transmitted through the sample. Recordings were performed using an amplifier (Molecular Devices, Multiclamp 700B), a digitizer (Molecular Devices, Digidata 1550B) at a sampling rate of 10 kHz and controlled using pCLAMP11 (Molecular Devices). During experimental sessions, slice cultures were perfused with artificial cerebrospinal fluid (ACSF) comprised of 125 mM NaCl, 2.5 mM KCl, 1.5 mM CaCl₂, 1 mM MgCl₂, 26 mM NaHCO₃, 0.3 mM ascorbic acid, 25 mM D-glucose, 1.25 mM NaH₂PO₄. Synaptic transmission was blocked during all experiments by the addition of 1 μM AP5 (Abcam, ab120003), 1 μM NBQX (Abcam, ab120046), and 10 μM picrotoxin (Abcam, ab120315) to the extracellular (recording) solution. Continuous aeration of the recording solution with 95% O₂ and 5% CO₂, resulted in a final pH of 7.4. Patch pipettes with a tip resistance of 4–6 MΩ were filled with an intracellular solution consisting of 135 mM K-gluconate, 4 mM KCl, 4 mM Mg-ATP, 0.3 mM Na-GTP, 10 mM Na₂-phosphocreatine, and 10 mM HEPES (pH 7.35). Only recordings with an access resistance below 30 MΩ were included in the subsequent analysis.

During experiments performed using whole-cell voltage clamp, neurons were held at −60 mV (the average resting potential of neurons in hippocampal organotypic slices). The soma of each patched neuron was precisely positioned in the center of the field of view. When recording the photocurrent as a function of membrane potential (holding potentials: −80, −70, −65, −60, −55 mV), neurons were temporarily held at each holding potential 5 s before and after photostimulation. For data presented in Fig. 5a–d, two-photon photoactivation was performed by continuous, 200 ms, illumination of each patched neuron using a 12-μm-diameter holographic spot (wavelengths: 850, 900, 920, 950, 980, 1000, 1050, 1100 nm), which was precisely positioned in the center of the field of view.

Data presented in Fig. 7d–g was acquired in current-clamp experiments. Where necessary, the current was injected to maintain neurons at the resting membrane potential (−60 mV).

The ability of two-photon holographic excitation to evoke action potentials was first assessed using a protocol consisting of 5, 5 ms pulses of 1100 nm light for power densities ranging between 0.16 and 1.00 mW μm⁻². The latency and jitter of light-evoked action potentials, respectively defined as the mean and standard deviation of the time between the onset of stimulation to the peak of the action potential, were measured using an identical protocol. Trains of light pulses with frequencies between (2–30 Hz) were used to verify that trains of action potentials could be reliably induced using 5 ms 1100 nm illumination.

The potency of two-photon inhibition was evaluated by measuring the rheobase shift induced by 920 nm illumination. The depolarizing current was injected for 5 ms into recorded neurons (from 0 to 1.2 nA in steps of 20 pA). The protocol was stopped when action potentials were observed for 3 consecutive current steps. The rheobase was defined as the amount of current injected to evoke the first of these 3 action potentials. The rheobase shift was measured by repeating the protocol with co-incident, 5 ms, illumination of the neuron with a 920 nm holographic spot (power densities between 0.05 and 0.25 mW μm⁻²). Co-incident trains of light pulses (15 ms) and injected current (10 ms) with frequencies between (2–30 Hz) were used to verify that two-photon inhibition could precisely and reliably eliminate single spikes.

Sustained neuronal silencing by two-photon excitation of somBiPOLES under 920 nm illumination was characterized by continuously injecting current above the rheobase for 1 s. The protocol was repeated with 200 ms co-incident illumination using a 920 nm holographic spot (power densities between 0.05 and 0.3 mW μm⁻²).

Two-photon, bidirectional, control of single neurons was demonstrated by co-incident illumination of titrated 920 nm and 1100 nm light. A 10-Hz train of 15 ms pulses of 1100 nm light was used to evoke a train of action potentials which were shunted using a continuous 200 ms pulse of 920 nm light.

**Two-photon photostimulation of somBiPOLES in hippocampal organotypic slices**. Two-photon photostimulation was performed using a tunable femtosecond laser (Coherent Discovery, 80 MHz, 100 fs, tuned between 850 and 1100 nm). A schematic diagram of the experimental setup is presented in Supplementary Fig. 10. A telescope formed of two lenses (L1 (Thorlabs, AC508-100-B) and L2 (Thorlabs, AC508-400-B)) expanded the beam onto a Spatial Light Modulator (SLM, Hamamatsu, LCOS 10468-07, 600 × 800 pixels, 20 μm pitch). In the schematic diagram, the reflective SLM is shown as transmissive for illustrative purposes. The SLM, controlled using custom-built software[62], was used to modulate the phase of the beam. Holograms designed to generate 12 μm holographic spots at the focal plane of the microscope were computed using an iterative Gerchberg-Saxton algorithm[63]. The zeroth diffraction order from the SLM was removed using a physical beam block. The modulated field was relayed and de-magnified using a pair of telescopes (formed of lenses L3 (Thorlabs, AC508-750-B), L4 (Thorlabs, AC508-750-B), L5 (Thorlabs, AC508-500-B) and L6 (Thorlabs, AC508-300-B)) to fill the back-aperture of the microscope objective (Nikon, CFI APO NIR, ×40, 0.8 NA) which projected the holograms onto the focal plane. Phase masks were calculated such that holographic spots for the light of different wavelengths overlapped laterally and axially. The anti-reflective coating of the lenses used are optimized for wavelengths 650–1050 nm, and losses incurred at 1100 nm result in the system being power limited at this wavelength. Hence, spectral characterization was performed by normalizing the power density at all wavelengths to the maximum transmitted at 1100 nm. The power incident on the sample plane was adjusted using a high-speed modulator (Thorlabs, OM6NH/M), which was calibrated for each experimental session for each wavelength used, to ensure a photon flux of 6.77 × 10²⁶ photons s⁻¹ m⁻² for all data presented in Fig. 5a. All powers were measured in the object plane using a power meter (Thorlabs, S121C). This experimental configuration was used for all data presented in Fig. 5a, along with all data acquired using 1100 nm illumination. Two-photon inhibition was performed using a femtosecond laser with a fixed wavelength (Spark Alcor, 80 MHz, 100 fs, 920 nm) which was combined with the beam from the tunable laser using a dichroic mirror (Thorlabs, DMLP950R). A liquid crystal variable retarder (Thorlabs, LCC1111-B) and a polarizing beam splitter (Thorlabs, PBS253) were combined to modulate the maximum power of the fixed 920 nm beam independently of that of the tunable laser. The power densities used in each experiment are specified alongside the relevant data in Fig. 5 and Supplementary Fig. 10.

**Transgenic C. elegans lines and transgenes**. The strain ZX417 (zxEx34[punc17::NpHR-ECFP;punc17::CHOP-2(H134R)::eYFP;rol-6]) was generated by injection of plasmid DNA (plasmids pRF4 (rol-6d), punc-17::NpHR-eCFP, and punc-17::ChR2 (H134R)-eYFP; each at 80 ng/μl) into the germline of C. elegans wild-type hermaphrodites. Transgenic animals were picked from the F1 generation and one line (ZX417) was selected out of several transgenic F2 lines for further experiments[33]. For expression in cholinergic neurons of C. elegans, BiPOLES (GtACR2::ts::mCerulean3::βHK::Chrimson) was subcloned into the punc-17 vector RM#348p (a gift from Jim Rand) via Gibson Assembly based on the plasmid CMV_GtACR2_mCerulean_βHK_Chrimson, using the restriction enzyme NheI and the primers ACR2_Chrimson_fwd (5′-atttcaggaggacccttggATGGCATCACAGGTCGTC-3′) and ACR2_Chrimson_rev (5′-ataccatggtaccgtcgacgTCACACTGTGTCCTCGTC-3′), resulting in the construct

pAB26. The respective transgenic strain ZX2586 (wild type; *zxEx1228[punc-17::GtACR2::ts::mCerulean3::βHK::Chrimson; pelt-2::GFP]*), was generated via microinjection[64] of both 30 ng μl⁻¹ plasmid and co-marker plasmid DNA p*elt-2*::GFP. Animals were cultivated on nematode growth medium (NGM), seeded with *E. coli* OP-50 strain, in 6 cm Petri dishes. To obtain functional rhodopsins in optogenetic experiments, the OP-50 bacteria were supplemented with all-*trans*-retinal ATR (0.25 μl of a 100 mM stock (in ethanol) mixed with 250 μl OP-50 bacterial suspension).

**C. elegans stimulation and behavioral experiments.** For body-length measurements, L4 stage transgenic animals were cultivated on ATR plates overnight. Video analysis of light-stimulation protocols provided information on depolarized and hyperpolarized states, based on contracted or relaxed body-wall muscles (BWMs)[65]. Prior to experiments, animals were singled on plain NGM plates to avoid imaging artefacts. They were manually tracked with an Axio Scope.A1 microscope (Zeiss, Germany), using a ×10 objective (Zeiss A-Plan 10x/0,25 Ph1 M27) and a Powershot G9 digital camera (Canon, USA). For light-stimulation of optogenetic tools, transgenic worms were illuminated with 5 s light pulses at 1.1 mW mm⁻² of different wavelengths as indicated in Fig. 6d (monochromatic light source, Polychrome V, Till Photonics or 100 W HBO mercury lamp with 470/40 ET Bandpass or 575/40 ET Bandpass filters, AHF Analysentechnik), controlled via an Arduino-driven shutter (Sutter Instrument, USA). Videos were processed and analyzed using a custom-written MATLAB script[66] (MathWorks, USA). For the analysis of data, the animals' body length was normalized to the recording period prior to illumination.

**Transgenic D. melanogaster lines and transgenes.** BiPOLES-mCerulean cDNA was cloned via blunt-end ligation into pJFRC7[67]. BILOES was cut with BamHI/HindIII and the vector was cut with NotI/XbaI. A transgenic line inserted into the attP2 site on the 3rd chromosome[68] was generated by phiC31-mediated site-specific transgenesis (FlyORF Injection Service, Zurich, Switzerland). A Gal4 line expressing in glutamatergic neurons including motor neurons (OK371-Gal4[11]) was used for locomotion experiments, a Dp7-expressing line (Ilp7-Gal4[34]) was used for mechanonociception experiments.

**Locomotion and mechanonociception assays in D. melanogaster larvae.** *D. melanogaster* larvae were staged in darkness on grape agar plates and fed with yeast paste containing 5 mM all-trans-retinal. Third instar larvae (96 h ± 2 h after egg laying) were used for all experiments.

For locomotion and body length analyses, animals were carefully transferred under minimum red light conditions to a 2% agar film on an FTIR (frustrated total internal reflection) based tracking system (FIM, University of Münster)[69]. Five freely moving larvae/trials were video-captured and stimulated with 470 nm (17 μW mm⁻²) or 635 nm (25 μW mm⁻²) light (CoolLED PE4000) for activation of BiPOLES. Animal locomotion was tracked with 10 frames per s for up to 70 s and then body length was analyzed using the FIMtracking software (FIM, University of Münster). For analysis, only animals displaying continuous locomotion before the light stimulus were kept. Larval body length was analyzed over time and was displayed with a 1 s moving average. The body length was normalized to the average of the first 5 s of recording. Relative body length changes during the experiment were then analyzed and plotted.

For mechanonociception, staged larvae were placed on 2% agar plates with a 1 ml water film added. Experiments were performed under minimum light conditions (no activation) with calibrated von-Frey-filaments (50 mN). For activation of BiPOLES, larvae were illuminated during the assay with either 470 nm (17 μW mm⁻²) or 635 nm (25 μW mm⁻²). Larvae were stimulated twice on mid-abdominal segments (a3–a6) within 2 s. Behavioral responses (stop and turning, bending, rolling) were noted, analyzed, and plotted. Staging and experiments were done in a blinded and randomized fashion.

**Modulation of noradrenergic neurons in the mouse locus coeruleus**

*Animals.* All procedures were in agreement with the German national animal care guidelines and approved by the Hamburg state authority for animal welfare (Behörde für Justiz und Verbraucherschutz) and the animal welfare officer of the University Medical Center Hamburg-Eppendorf. Experiments were performed on mice of either sex between 2.5 and 4 months of age at the start of the experiment. Mice were obtained from The Jackson Laboratory, bred, and maintained at our own colony (12/12 h light-dark cycle, 22 °C room temperature, ~40% relative humidity, food, and water ad libitum). Transgenic mice expressing Cre recombinase in tyrosine hydroxylase positive neurons (TH-Cre, Stock No: 008601)[70] were injected with a suspension of rAAV2/9 viral particles encoding hSyn-DIO-somBiPOLES to target Cre-expressing neurons in the locus coeruleus. Control experiments were performed in non-injected wild-type littermates.

*Virus injection and implantation of optic fibers.* General anesthesia and analgesia were achieved by intraperitoneal injections of midazolam/medetomidine/fentanyl (5.0/0.5/0.05 mg kg⁻¹, diluted in NaCl). After confirming anesthesia and analgesia by the absence of the hind limb withdrawal reflex, the scalp of the animal was trimmed and disinfected with Iodide solution (Betaisodona; Mundipharma, Germany). The animal was placed on a heating pad to maintain body temperature,

fixed in a stereotactic frame, and eye ointment (Vidisic; Bausch + Lomb, Germany) was applied to prevent drying of the eyes. To bilaterally access the LC, an incision (~1 cm) was made along the midline of the scalp, the skull was cleaned, and small craniotomies were drilled −5.4 mm posterior and ±1 mm lateral to Bregma. 0.4 μl of virus suspension were injected into each LC (−3.6 mm relative to Bregma) at a speed of ~100–200 nl min⁻¹ using a custom-made air pressure system connected to a glass micropipette. After each injection, the micropipette was left in place for a minimum of 5 min before removal. After virus injection, cannulas housing two ferrule-coupled optical fibers (200 μm core diameter, 0.37 NA, 4 mm length) spaced 2 mm apart (TFC_200/245-0.37_4mm_TS2.0_FLT; Doric Lenses, Canada) were inserted just above the injection site to a depth of −3.5 mm relative to Bregma using a stereotactic micromanipulator. The implant, as well as a headpost for animal fixation during the experiment, were fixed to the roughened skull using cyanoacrylate glue (Pattex; Henkel, Germany) and dental cement (Super Bond C&B; Sun Medical, Japan). The incised skin was glued to the cement to close the wound. Anesthesia was antagonized by intraperitoneally injecting a cocktail of atipamezole/flumazenil/buprenorphine (2.5/0.5/0.1 mg kg⁻¹, diluted in NaCl). Carprofen (4 mg kg⁻¹) was given subcutaneously for additional analgesia and to avoid inflammation. In addition, animals received meloxicam mixed into softened food for 3 days after surgery.

*Optogenetic stimulation.* Four to six weeks after surgery, mice were habituated to head fixation and placement in a movement-restraining plastic tube for at least one session. Bilateral optogenetic stimulation of LC neurons was achieved by connecting the fiber implant to a 1 × 2 step-index multimode fiber optic coupler (200 μm core diameter, 0.39 NA; TT200SL1A, Thorlabs, Germany) in turn connected to a laser combiner system (LightHUB; Omicron, Germany) housing a 473 nm (LuxX 473-100; Omicron, Germany) and a 594 nm diode laser (Obis 594 nm LS 100 mW; Coherent, Germany) for activation of the *Gt*ACR2 and Chrimson components of somBiPOLES, respectively. Coupling to the implant was achieved with zirconia mating sleeves (SLEEVE_ZR_1.25; Doric lenses, Canada) wrapped with black tape to avoid light emission from the coupling interface. Following a habituation period of ~3 min after placing mice in the setup, stimuli were generated and presented using custom-written MATLAB scripts (MathWorks, US) controlling a NI-DAQ-card (PCIe-6323; National Instruments, US) to trigger the lasers via digital input channels. For activation of Chrimson, pulse trains (594 nm, ~10 mW at each fiber end, 20 ms pulse duration, 20 Hz repetition rate) of 4 s duration were presented, while *Gt*ACR2 was activated by continuous illumination (473 nm, ~10 mW at each fiber end) of 2–6 s duration. 30–40 trials of 473 nm pulses, 594 nm pulse trains, and combinations thereof, were presented at an inter-train-interval of 20–30 s in each session.

*Data acquisition.* A monochrome camera (DMK 33UX249; The Imaging Source, Germany) equipped with a macro objective (TMN 1.0/50; The Imaging Source, Germany) and a 780 nm long-pass filter (FGL780; Thorlabs, Germany) was pointed towards one eye of the mouse. Background illumination was provided with an infrared spotlight (850 nm), while a UV LED (395 nm; Nichia, Japan) was adjusted to maintain pupil dilation of the mouse at a moderate baseline level. Single frames were triggered at 30 Hz by an additional channel of the NI-DAQ-card that controlled optogenetic stimulation, and synchronization was achieved by simultaneous recording of all control voltages and their corresponding timestamps.

*Data analysis.* Pupil diameter was estimated using a custom-modified, MATLAB-based algorithm developed by McGinley et al.[71]. In short, an intensity threshold was chosen for each recording to roughly separate between pupil (dark) and non-pupil (bright) pixels. For each frame, a circle around the center of mass of putative pupil pixels and with an area equivalent to the number of pupil pixels was then calculated, and putative edge pixels were identified by canny edge detection. Putative edge pixels that were more than 3 pixels away from pixels below the threshold (putative pupil) or outside an area of ±0.25–1.5 times the diameter of the fitted circle were neglected. Using least-squares regression, an ellipse was then fit on the remaining edge pixels, and the diameter of a circle of the equivalent area to this ellipse was taken as the pupil diameter. Noisy frames (e.g., no visible pupil due to blinking or blurry pupil images due to saccades of the animal) were linearly interpolated, and the data was low-passed filtered (<3 Hz; 3rd order Butterworth filter). Pupil data was segmented from 5 s before to 15 s after the onset of each stimulus and normalized to the median pupil diameter of the 5 s preceding the stimulus onset, before individual trials were averaged. Randomly chosen segments of pupil data of the same duration served as a control. The difference in median pupil diameter one second before and after stimulation (as indicated in Fig. 7c) was used to calculate potential changes in pupil diameter for each condition. Statistical significance was calculated using one-way analysis of variance and Tukey's post-hoc multiple comparison tests.

**In-vivo recordings from ferret visual cortex.** Data were collected from 3 adult female ferrets (*Mustela putorius*). All experiments were approved by the independent Hamburg state authority for animal welfare (Behörde für Justiz und Verbraucherschutz) and were performed in accordance with the guidelines of the German Animal Protection Law and the animal welfare officer of the University Medical Center Hamburg-Eppendorf.

**Table 2 Photon flux given as a number of photons s⁻¹ m⁻².**

| | Irradiance (mW mm⁻²) | | | | | |
|---|---|---|---|---|---|---|
| | 0.001 | 0.01 | 0.1 | 1 | 10 | 100 |
| Wavelength (nm) | | | | | | |
| 365 | 1.84E + 18 | 1.84E + 19 | 1.84E + 20 | 1.84E + 21 | 1.84E + 22 | 1.84E + 23 |
| 385 | 1.94E + 18 | 1.94E + 19 | 1.94E + 20 | 1.94E + 21 | 1.94E + 22 | 1.94E + 23 |
| 405 | 2.04E + 18 | 2.04E + 19 | 2.04E + 20 | 2.04E + 21 | 2.04E + 22 | 2.04E + 23 |
| 435 | 2.19E + 18 | 2.19E + 19 | 2.19E + 20 | 2.19E + 21 | 2.19E + 22 | 2.19E + 23 |
| 460 | 2.32E + 18 | 2.32E + 19 | 2.32E + 20 | 2.32E + 21 | 2.32E + 22 | 2.32E + 23 |
| 470 | 2.37E + 18 | 2.37E + 19 | 2.37E + 20 | 2.37E + 21 | 2.37E + 22 | 2.37E + 23 |
| 490 | 2.47E + 18 | 2.47E + 19 | 2.47E + 20 | 2.47E + 21 | 2.47E + 22 | 2.47E + 23 |
| 525 | 2.65E + 18 | 2.65E + 19 | 2.65E + 20 | 2.65E + 21 | 2.65E + 22 | 2.65E + 23 |
| 550 | 2.77E + 18 | 2.77E + 19 | 2.77E + 20 | 2.77E + 21 | 2.77E + 22 | 2.77E + 23 |
| 580 | 2.92E + 18 | 2.92E + 19 | 2.92E + 20 | 2.92E + 21 | 2.92E + 22 | 2.92E +;23 |
| 595 | 3E + 18 | 3E + 19 | 3E + 20 | 3E + 21 | 3E + 22 | 3E + 23 |
| 630 | 3.18E + 18 | 3.18E + 19 | 3.18E + 20 | 3.18E + 21 | 3.18E + 22 | 3.18E + 23 |
| 660 | 3.33E + 18 | 3.33E + 19 | 3.33E + 20 | 3.33E + 21 | 3.33E + 22 | 3.33E + 23 |

For injection of rAAV2/9 viral particles encoding mDlx-BiPOLES-mCerulean (see Table 2) animals were anesthetized with an injection of ketamine (15 mg kg⁻¹), medetomidine (0.02 mg kg⁻¹), midazolam (0.5 mg kg⁻¹) and atropine (0.15 mg kg⁻¹). Subsequently, they were intubated and respiration with a mixture of 70:30 $N_2/O_2$ and 1–1.5% isoflurane. A cannula was inserted into the femoral vein to deliver a bolus injection of enrofloxacin (15 mg kg⁻¹) and Rimadyl (4 mg kg⁻¹) and, subsequently, continuous infusion of 0.9% NaCl and fentanyl (0.01 mg kg⁻¹ h⁻¹). Body temperature, heart rate, and end-tidal $CO_2$ were constantly monitored throughout the surgery. Before fixing the animal's head in the stereotaxic frame, a local anesthetic (Lidocaine, 10%) was applied to the external auditory canal. The temporalis muscle was folded back, such that a small craniotomy (ø: 2.5 mm) could be performed over the left posterior cortex and the viral construct was slowly (0.1 µl min⁻¹) injected into the secondary visual cortex (area 18). The excised piece of bone was put back in place and fixed with tissue-safe silicone (Kwikcast; WPI). Also, the temporalis muscle was returned to its physiological position and the skin was closed. After the surgery, the animals received preventive analgesics (Metacam, 0.1 mg) and antibiotics (Enrofloxacin, 15 mg kg⁻¹) for ten days.

After an expression period of at least 4 weeks, recordings of cortical signals were carried out under isoflurane anesthesia. Anesthesia induction and maintenance were similar to the procedures described above, except for a tracheotomy performed to allow for artificial ventilation of the animal over an extended period. The i.v. infusion was supplemented with pancuronium bromide (6 µg kg⁻¹ h⁻¹) to prevent slow ocular drifts. To keep the animal's head in a stable position throughout the placement of recording electrodes and the measurements, a headpost was fixed with screws and dental acrylic to the frontal bone of the head. Again, the temporalis muscle was folded back and a portion of the cranial bone was resected. The dura was removed before introducing an optrode with 32 linearly distributed electrodes (A1x32-15mm-50(100)-177, NeuroNexus Technologies) into the former virus-injection site (area 18). The optrode was manually advanced via a micromanipulator (David Kopf Instruments) under visual inspection until the optic fiber was positioned above the pial surface and the uppermost electrode caught a physiological signal, indicating that it had just entered the cortex.

During electrophysiological recordings, the isoflurane level was maintained at 0.7%. To ensure controlled conditions for sensory stimulation, all experiments were carried out in a dark, sound-attenuated anechoic chamber (Acoustair, Moerkapelle, Netherlands). Visual stimuli were created via an LED placed in front of the animal's eye. In separate blocks, 150 laser stimuli of different colors ('red', 633 nm LuxXplus and 'blue', 473 nm LuxXplus, LightHub-4, Omicron) were applied through the optrode for 500 ms, each, at a variable interval of 2.5–3 s. Randomly, 75 laser stimuli were accompanied by a 10 ms LED flash, starting 100 ms after the respective laser onset. For control, one block of 75 LED flashes alone was presented at comparable interstimulus intervals.

Electrophysiological signals were sampled with an AlphaLab SnR recording system (Alpha Omega Engineering, Nazareth, Israel) or with a self-developed neural recording system based on INTAN digital head-stages (RHD2132, Intantech). Signals recorded from the intracortical laminar probe were band-pass filtered between 0.5 Hz and 7.5 kHz and digitized at 22–44 kHz or 25 kHz, respectively. All analyses of neural data presented in this study were performed offline after the completion of experiments using MATLAB scripts (MathWorks). To extract multiunit spiking activity (MUA) from broadband extracellular recordings, we high-pass filtered signals at 500 Hz and detected spikes at negative threshold (>3.5 SD)[72].

**Reporting summary**. Further information on research design is available in the Nature Research Reporting Summary linked to this article.

## Data availability
Source data are provided with this paper. All data generated in this study are provided in the Source Data file. Source data are provided with this paper.

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

## Acknowledgements

We thank Stefan Schillemeit and Tharsana Tharmalingam for excellent technical assistance, Mathew McGinley and Peter Murphy for help with pupil analysis, and Sonja Kleinlogel for providing plasmids carrying the original opsin tandem. We also thank Karl Deisseroth and Charu Ramakrishnan for providing the plasmids and coding sequences of bReaChES and eNPAC2.0, as well as for providing the ChRmine plasmid and coding sequence in advance of publication. We further thank Jonas Wietek for providing ACR plasmids and for discussions at an early phase of the project. Ingke Braren of the UKE Vector Facility produced AAV vectors. This work was supported by the German Research Foundation, DFG (SPP1926, FOR2419/P6, SFB936/B8 to J.S.W., SFB936/A2 and SPP2041/EN533/15-1 to A.K.E., SPP1926 and SFB1315 to P.H., SFB807/P11 to A.C.F.B. and A.G.), the 'Agence Nationale de la Recherche' (CE16-2019 HOLOPTOGEN, CE16-0021 SLALLOM, ANR-10-LABX-65 LabEx LIFESENSES, and ANR-18-IAHU-01 *IHU FOReSIGHT to V.E.), the AXA research foundation and the European Research Council (ERC2016-StG-714762 to J.S.W., HOLOVIS-AdG to V.E., Stardust H2020 767092 to P.H.). Peter Hegemann is a Hertie Professor and supported by the Hertie Foundation.

## Author contributions

Conceptualization: J.V., S.R.R., P.H., J.S.W.; investigation: J.V., S.R.R., A.D., F.P., R.S., F.T., A.C.F.B., I.B., F.Z., N.Z., J.A., S.A., K.S., J.S.W.; data curation: J.V., S.R.R., A.D., F.P.,

R.S., F.T., A.C.F.B., J.A.; analysis: J.V., S.R.R., A.D., F.P., R.S., F.T., A.C.F.B., I.B., F.Z., J.A., J.S.W.; software: A.D., F.P.; supervision: E.P., A.G., P.S., V.E., A.K.E., P.H., J.S.W.; funding acquisition: E.P., A.G., P.S., V.E., A.K.E., P.H., J.S.W.; project administration: P.H., J.S.W.; writing: J.V., S.R.R., F.P., R.S., J.S.W. with contributions from all authors.

## Funding

## Competing interests
The authors declare no competing interests.
