## [Peer Review File · Nature Communications]

Reviewers' Comments:

Reviewer #1:

Remarks to the Author:

The authors present the development, characterisation, and many applications of a novel optogenetic tool for bidirectional excitation and inhibition in the same neurons. This is a much needed advance in the optogenetic toolbox as the development of crucial bidirectional tools has stagnated since their initial development. The authors cleverly use a recently discovered strong inhibitory channel to circumvent the blue light sensitivity, from which all red light based tools suffer. The suite of applications enabled by this tool is impressively large and shown via rigorously controlled experiments. This tool is not yet perfect however it is an important step forward in the development of tools of this type. I support publication of this manuscript in its present form with only very minor edits below.

Minor comments:

1. In the introduction, the authors mention in different places some requirements for generating an appropriate tool of this type. On line 33 they mention "matched photocurrent amplitudes" and on line 53 "equal sub cellular distributions". These might be required under certain experimental constraints but not all. It might be productive to further discuss the logic of these reasons, should space allow it.

2. Related to Fig 2c: It might be useful (but not necessary) to also see an image and/or comparison of fluorescence levels out in more distal dendrites, if the authors might have such images already available.

Very minor comments:

Line 197: missing comma after "cultures"

Line 204: should "eviting" possibly be "evicting"?

Line 168 of supplement: tuneable -> tunable

Fig 5a: The trace "colors" (black->grey) are swapped since Fig 1, which is ever so slightly confusing.

Reviewer #2:

Remarks to the Author:

The authors have made several constructs encoding a neuron activator fused to a neuron inhibitor to form a single polypeptide chain capable of activation or inhibition of neuron spiking depending on wavelength of light. The constructs are based on the first dual-color optogenetic tools developed by Ernst Bamberg's laboratory in 2011 (author's reference 2). Bamberg's constructs fused the cation-conducting rhodopsin channel (CCR) Channelrhodopsin-2 (ChR2) and its mutants with hyperpolarizing ion pumps such as the chloride pump halorhodopsin or proton pump archaerhodopsin. By joining the excitatory ChR2 with the inhibitory ion pumps, Bamberg's group demonstrated light-regulated quantitative color-specific bi-directional control of the membrane potential in HEK293 cells and neurons in vitro, but as optogenetic tools the molecules suffered primarily from the relatively weak inhibition from light-driven ion pumps, especially in the fused constructs.

The authors have extended the strategy of Bamberg by taking advantage of improvements in optogenetic tools over this past decade. They found one combination that produces an effective construct, the GtACR2-L2-Chrimson pair (L2 = linker). GtACR2 is a blue-absorbing anion-conducting channelrhodopsin orders of magnitude more effective than any other neuron firing inhibitor (except GtACR1, which has similar potency but is green-absorbing), and Chrimson is a far red-shifted CCR, providing large spectral distance from the blue-absorbing inhibitor. The construct designated a "BIPOLE" is much improved over earlier constructs. The authors also added a soma-dendritic targeting peptide previously found to be important to avoid unwanted axonal expression of GtACR2 and frequently used with GtACRs in model animals. This worked well also for the fusion

construct keeping it out of the axon region.

The authors further demonstrate that the GtACR2-L2-Chrimson pair works well in living animal model systems *C. elegans*, *Drosophila*, mice and ferrets. GtACRs have been successfully used in each of these systems, but only as an isolated inhibitor.

The technical work is expertly performed throughout, the data clearly presented, and the manuscript concisely and clearly written in most parts. However, one part in Introduction needs to be modified to avoid misleading the reader:

The 2 crucial discoveries that made possible the successful development of an effective BIPOLE are (1) the far red absorption spectrum of Chrimson and (2) the orders-of-magnitude greater conductance of GtACR2 compared to other channelrhodopsins. These findings are confirmed in this work, but they are not new findings in this paper. These facts will not be clear to the reader because there is no reference to the discovery papers at both proteins' first mention in the Introduction "...Among all tested variants, a combination of GtACR2 and Chrimson termed BiPOLES ... (beginning line 97). References to the laboratories that discovered these proteins should go here: Ed Boyden's lab for Chrimson (Ref 12), and John Spudich's laboratory for GtACR2 (Ref 19).

Reviewer #3:

Remarks to the Author:

The manuscript reports the development and characterization of an optogenetic tool called BiPOLES. As for tool development work, the report scores in all critical categories. The concept of this tool, the activation and silencing of the same neuron with light of different wavelengths, is not new. However, current versions of dual-color bidirectional optogenetic constructs have specific limitations that the new tool resolves. These are 1:1 membrane-localized co-expression of the activating and silencing components by designing and testing β HK-based fusion constructs, utility in organisms where pump-based tools are inefficient by utilizing ion channels for both components, and bi-directional control of neuronal activity over a range of light intensities by inversion of the previously applied blue-light activation and red-light inhibition. The investigators provide extensive experimental data on the development of the best candidate molecule (a combination of GtACR2 and Chrimson, i.e. BiPOLES, further improved by somatic targeting, i.e. somBiPOLES) and its biophysical characterization as well as benchmarking the new tool against the most utilized current tool (eNPAC2.0). Most importantly, the authors demonstrate convincingly that somBiPOLES allows bidirectional optogenetic control by both single- or two-photon excitation, the latter in single neurons, and that the tool performs efficiently in vivo in behavioral paradigms in *C. elegans*, *Drosophila*, mice and ferrets.

The introduced tool is not simply an improved version of previous constructs but offers features, based on its design that cannot be realized with those currently available. This particularly pertains to the option of membrane tuning taking advantage of the combination of anion and cation channels, and by enabling the use of the many available blue-light activated excitatory opsins for dual color activation (blue with red excitation from BiPOLES) of two intermingled, but distinct neuronal populations without any cross-talk.

The results of the paper will be of wide interest to the neuroscience community as well as to the larger optogenetic and photoreceptor communities. The reported results are extensive, of high quality and convincing. The details provided in the Methods section will allow others to reproduce the work and to use the constructs for their experiments "out-of-the-box".

Two minor comments:

The authors show that both BiPOLES and somBiPOLES work in vivo, as the experiments in worms, flies, and ferrets were done with BiPOLES, and those in mice with somBiPOLES. The assumption is that this was done along a historical timeline, with BiPOLES available before somBiPOLES. Nevertheless, in the discussion the authors might mention if there are specific indications/contraindications for the use of BiPOLES versus somBiPOLES.

main text, line 204: targeting has the additional benefit of eviting expression of the construct in axon terminals – replace “eviting” by “avoiding”

REVIEWER COMMENTS

We are delighted that all three referees acknowledge the quality and significance of our work. We thank them for their careful evaluation of our manuscript and for their positive comments. We addressed all remaining issues in the revised manuscript and we are looking forward to seeing it published, soon. Our responses to the individual points raised by the reviewers can be found below.

Reviewer #1 (Remarks to the Author):

The authors present the development, characterisation, and many applications of a novel optogenetic tool for bidirectional excitation and inhibition in the same neurons. This is a much needed advance in the optogenetic toolbox as the development of crucial bidirectional tools has stagnated since their initial development. The authors cleverly use a recently discovered strong inhibitory channel to circumvent the blue light sensitivity, from which all red light based tools suffer. The suite of applications enabled by this tool is impressively large and shown via rigorously controlled experiments. This tool is not yet perfect however it is an important step forward in the development of tools of this type. I support publication of this manuscript in its present form with only very minor edits below.

Minor comments:

1. In the introduction, the authors mention in different places some requirements for generating an appropriate tool of this type. On line 33 they mention "matched photocurrent amplitudes" and on line 53 "equal sub cellular distributions". These might be required under certain experimental constraints but not all. It might be productive to further discuss the logic of these reasons, should space allow it.

We expanded our discussion to better explain what we mean by "matched photocurrent amplitudes" (lines 603-608) and to encompass some experimental conditions where "equal sub cellular distributions" of both opsins are desirable (lines 685-694).

2. Related to Fig 2c: It might be useful (but not necessary) to also see an image and/or comparison of fluorescence levels out in more distal dendrites, if the authors might have such images already available.

We added example images of CA3 neurons in stratum oriens expressing either BiPOLES or somBiPOLES to supplemental fig. 3.

Very minor comments:

Line 197: missing comma after "cultures"

Line 204: should "eviting" possibly be "evicting"?

Line 168 of supplement: tuneable -> tunable

Fig 5a: The trace "colors" (black->grey) are swapped since Fig 1, which is ever so slightly confusing.

fixed

Reviewer #2 (Remarks to the Author):

The authors have made several constructs encoding a neuron activator fused to a neuron inhibitor to form a single polypeptide chain capable of activation or inhibition of neuron spiking depending on wavelength of light. The constructs are based on the first dual-color optogenetic tools developed by

Ernst Bamberg's laboratory in 2011 (author's reference 2). Bamberg's constructs fused the cation-conducting rhodopsin channel (CCR) Channelrhodopsin-2 (ChR2) and its mutants with hyperpolarizing ion pumps such as the chloride pump halorhodopsin or proton pump archaerhodopsin. By joining the excitatory ChR2 with the inhibitory ion pumps, Bamberg's group demonstrated light-regulated quantitative color-specific bi-directional control of the membrane potential in HEK293 cells and neurons in vitro, but as optogenetic tools the molecules suffered primarily from the relatively weak inhibition from light-driven ion pumps, especially in the fused constructs.

The authors have extended the strategy of Bamberg by taking advantage of improvements in optogenetic tools over this past decade. They found one combination that produces an effective construct, the GtACR2-L2-Chrimson pair (L2 = linker). GtACR2 is a blue-absorbing anion-conducting channelrhodopsin orders of magnitude more effective than any other neuron firing inhibitor (except GtACR1, which has similar potency but is green-absorbing), and Chrimson is a far red-shifted CCR, providing large spectral distance from the blue-absorbing inhibitor. The construct designated a "BIPOLE" is much improved over earlier constructs. The authors also added a soma-dendritic targeting peptide previously found to be important to avoid unwanted axonal expression of GtACR2 and frequently used with GtACRs in model animals. This worked well also for the fusion construct keeping it out of the axon region.

The authors further demonstrate that the GtACR2-L2-Chrimson pair works well in living animal model systems *C. elegans*, *Drosophila*, mice and ferrets. GtACRs have been successfully used in each of these systems, but only as an isolated inhibitor.

The technical work is expertly performed throughout, the data clearly presented, and the manuscript concisely and clearly written in most parts. However, one part in Introduction needs to be modified to avoid misleading the reader:

The 2 crucial discoveries that made possible the successful development of an effective BIPOLE are (1) the far red absorption spectrum of Chrimson and (2) the orders-of-magnitude greater conductance of GtACR2 compared to other channelrhodopsins. These findings are confirmed in this work, but they are not new findings in this paper. These facts will not be clear to the reader because there is no reference to the discovery papers at both proteins' first mention in the Introduction "...Among all tested variants, a combination of GtACR2 and Chrimson termed BiPOLES ... (beginning line 97). References to the laboratories that discovered these proteins should go here: Ed Boyden's lab for Chrimson (Ref 12), and John Spudich's laboratory for GtACR2 (Ref 19).

We apologize for the oversight and inserted the references as suggested by the reviewer.

Reviewer #3 (Remarks to the Author):

The manuscript reports the development and characterization of an optogenetic tool called BiPOLES. As for tool development work, the report scores in all critical categories. The concept of this tool, the activation and silencing of the same neuron with light of different wavelengths, is not new. However, current versions of dual-color bidirectional optogenetic constructs have specific limitations that the new tool resolves. These are 1:1 membrane-localized co-expression of the activating and silencing components by designing and testing β HK-based fusion constructs, utility in organisms where pump-based tools are inefficient by utilizing ion channels for both components, and bi-directional control of neuronal activity over a range of light intensities by inversion of the previously applied blue-light activation and red-light inhibition. The investigators provide extensive experimental data on the development of the best candidate molecule (a combination of GtACR2 and Chrimson, i.e. BiPOLES, further improved by somatic targeting, i.e. somBiPOLES) and its biophysical

characterization as well as benchmarking the new tool against the most utilized current tool (eNPAC2.0). Most importantly, the authors demonstrate convincingly that somBiPOLES allows bidirectional optogenetic control by both single- or two-photon excitation, the latter in single neurons, and that the tool performs efficiently in vivo in behavioral paradigms in *C. elegans*, *Drosophila*, mice and ferrets.

The introduced tool is not simply an improved version of previous constructs but offers features, based on its design that cannot be realized with those currently available. This particularly pertains to the option of membrane tuning taking advantage of the combination of anion and cation channels, and by enabling the use of the many available blue-light activated excitatory opsins for dual color activation (blue with red excitation from BiPOLES) of two intermingled, but distinct neuronal populations without any cross-talk.

The results of the paper will be of wide interest to the neuroscience community as well as to the larger optogenetic and photoreceptor communities. The reported results are extensive, of high quality and convincing. The details provided in the Methods section will allow others to reproduce the work and to use the constructs for their experiments “out-of-the-box”.

Two minor comments:

The authors show that both BiPOLES and somBiPOLES work in vivo, as the experiments in worms, flies, and ferrets were done with BiPOLES, and those in mice with somBiPOLES. The assumption is that this was done along a historical timeline, with BiPOLES available before somBiPOLES. Nevertheless, in the discussion the authors might mention if there are specific indications/contraindications for the use of BiPOLES versus somBiPOLES.

We added a paragraph to the Discussion section explaining the rationale for the use of BiPOLES and somBiPOLES (lines 587-596).

main text, line 204: targeting has the additional benefit of eviting expression of the construct in axon terminals – replace “eviting” by “avoiding”

fixed

Ute Hochgeschwender